# KV-Dict: Sparse KV Cache Compression with Universal Dictionaries

## Abstract

Transformer has become the de facto architecture for Large Language Models (LLMs), yet its substantial memory required for long contexts makes it costly to deploy. Managing the memory usage of the key-value (KV) cache during inference has become a pressing challenge, as the cache grows with both model size and input length, consuming significant GPU memory.

We introduce a novel post-training KV cache compression method using KV-Dict, a universal dictionary that can accurately decompose and reconstruct key-value states. Unlike traditional quantization methods, KV-Dict leverages sparse dictionary learning, allowing for flexible memory usage with minimal performance loss through fine-grained controls of sparsity levels. Moreover, we retain competitive performance in the low memory regimes that 2-bit compression struggles to offer.

KV-Dict is remarkably universal, as it uses a small, input-agnostic dictionary that is shared across tasks and batches without scaling memory. This universality, combined with the ability to control sparsity for different memory requirements, offers a flexible and efficient solution to the KV cache bottleneck, maintaining strong performance on complex reasoning tasks, such as LongBench and GSM8K.

## 1 Introduction

Transformers (Vaswani et al., 2017) have become the backbone of frontier Large Language Models (LLMs), driving progress in domains beyond natural language processing. However, Transformers are often limited by their significant memory demands. This stems not only from the large number of model parameters, but also from the need to maintain the key-value (KV) cache that grows proportional to the model size and length of the input. Additionally, each input requires its own KV cache, limiting opportunities for reuse across different user inputs. This creates a bottleneck in generation speed for GPUs with limited memory (Yu et al., 2022) and thus, it has become crucial to alleviate KV cache memory usage while preserving its original performance across domains.

KV cache optimization research has explored both training-stage optimizations (Shazeer, 2019; Dai et al., 2024; Sun et al., 2024) and deployment-focused strategies (Kwon et al., 2023; Lin et al., 2024; Ye et al., 2024) to improve the efficiency of serving LLMs. Architectural approaches such as Grouped Query Attention (GQA) (Ainslie et al., 2023) reduce the number of KV heads, effectively reducing KV cache. However, these methods are not applicable as off-the-shelf methods to reduce KV cache for pretrained LLMs, leading to post-training compression efforts.

Post-training approaches include selectively retaining certain tokens (Beltagy et al., 2020; Xiao et al., 2023; Zhang et al., 2024) and quantization methods, which have had empirical success when quantizing KV cache into 2 or 4 bits (Liu et al., 2024b; He et al., 2024; Kang et al., 2024). However, eviction strategies have limitations on long-context tasks that require retaining a majority of previous tokens, while quantizations to 2 or 4 bits have clear upper bounds on compression rates.

In this paper, we focus on utilizing low-dimensional structures for efficient KV cache compression. Prior work reports that each key state lies in a low-rank subspace (Singhania et al., 2024; Wang et al., 2024b; Yu et al., 2024). Yet, it is unclear if *all states* lie in the same subspace; if so, such redundancy remains to be taken advantage of. Thus, we naturally ask the following questions:

> *Can we find substantial redundancy among keys and values, universal of all inputs?*
> *How can we leverage this for efficient KV cache compression?*

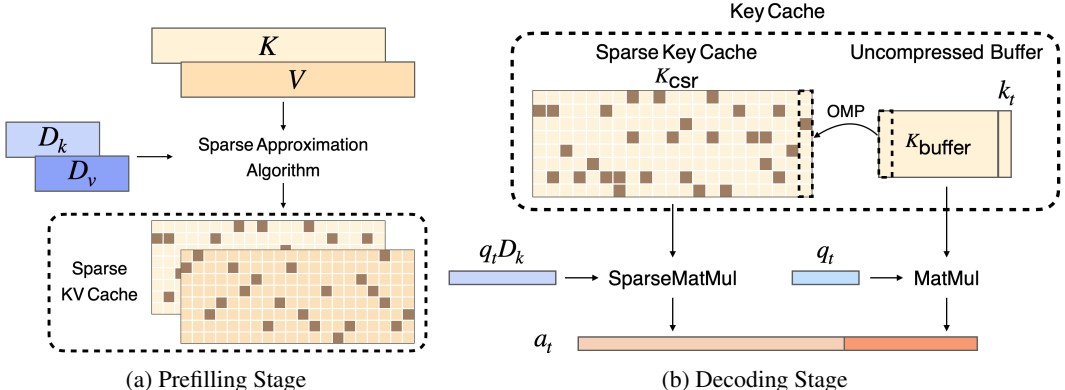

(a) Prefilling Stage  (b) Decoding Stage

Figure 1: **(a) Prefilling:** Following attention computation, KV-Dict uses OMP to find sparse representations of the KV states ($3$-$8\times$ smaller). **(b) Decoding:** Key cache consists of the compressed sparse key cache, $\boldsymbol{K}_{\text{csr}}$, and an full-precision buffer, $\boldsymbol{K}_{\text{buffer}}$, for the most recent tokens. $\boldsymbol{q}_t$, $\boldsymbol{k}_t$ represent the query, key vectors for the newly generated token. Computation is reduced by computing the query-dictionary product, $\boldsymbol{q}_t \boldsymbol{D}_k$, then multiplying $\boldsymbol{K}_{\text{csr}}$, to get the pre-softmax attention score.

Towards this end, we propose KV-Dict, a universal dictionary that serves as an overcomplete basis, which can sparsely decompose and reconstruct KV cache with sufficiently small reconstruction error. As shown in Section 2.2, we observe that a subset of key states cluster near each other, even though the keys are from different inputs, while some cluster on different subspaces. To accommodate for such low-dimensional structures, we use sparse dictionary learning, which has developed algorithms for information compression across various domains, such as signal processing and medical imaging (Candès et al., 2006; Donoho, 2006; Dong et al., 2014; Metzler et al., 2016).

KV-Dict is simple to learn, can be applied off-the-shelf for KV cache compression, and only occupies small constant memory regardless of input or batch size. Methodologically, KV-Dict utilizes both sparsity-based compression (steps 1 and 2) and quantization (step 3) in three straightforward steps:

1. **Dictionary pretraining:** As in Figure 3, we train a dictionary on WikiText-103 (Merity, 2016) for each model. This dictionary is only trained once and used universally across all tasks. It only occupies constant memory and does not increase with batch size.

2. **Sparse decomposition:** During prefilling and decoding (Figure 1), KV-Dict decomposes key-value into a sparse linear combination, which consists of $s$ pairs of sparse coefficient and dictionary index. This step by itself provides high compression rates.

3. **Lightweight sparse coefficients:** We obtain higher KV cache compression rates by representing the sparse coefficients in 8 bits instead of `FP16`. Lowering precision to 8 bits yields minimal degradation. KV-Dict theoretically allows us to compress more than 2-bit quantization ($1/8$ of `FP16` KV cache size) if $s \leq 10$ when head dimension is 128.

Overall, we make the following contributions:

- **Near-lossless performance:** Given similar memory requirements, KV-Dict performs on par with or better than baseline quantization methods on challenging language tasks, such as LongBench (Bai et al., 2023) and GSM8K (Cobbe et al., 2021).

- **Compression rates beyond 2-bits:** KV-Dict's sparsity parameter enables both wider and more fine-grained control over desired memory usage. This allows us to explore performance when using under 15-20% of the original KV cache size, a low-memory regime previous compression methods could not explore.

- **Universality:** Instead of an input-dependent dictionary, we find a sufficiently small universal dictionary (per model) that can be used for all tasks and across multiple users. Advantageously, such dictionary does *not* scale with batch size and can be used off-the-shelf.

## 2 KV CACHE COMPRESSION WITH DICTIONARIES

### 2.1 BACKGROUND & NOTATION

During autoregressive decoding in Transformer, the key and value states for preceding tokens are independent of subsequent tokens. As a result, these key and value states are cached to avoid recomputation, thereby accelerating the decoding process.

Let the input token embeddings be denoted as $X \in \mathbb{R}^{l_{seq} \times d}$, where $l_{seq}$ and $d$ are the sequence length and model hidden dimension, respectively. For simplicity, we focus on a single layer and express the computation of query, key, and value states at each attention head during the prefilling stage as:

$$Q^{(h)} = XW_q^{(h)}, \quad K^{(h)} = XW_k^{(h)}, \quad V^{(h)} = XW_v^{(h)}, \tag{1}$$

where $W_q^{(h)}, W_k^{(h)}, W_v^{(h)} \in \mathbb{R}^{d \times m}$ are the model weights with $m$ representing the head dimension.

Let $t$ represent the current step in the autoregressive decoding, and let $x_t \in \mathbb{R}^{1 \times d}$ denote the embedding of the newly generated token. The KV cache up to but not including the current token, are denoted as $K_{t-1}^{(h)}$ and $V_{t-1}^{(h)}$, respectively. The typical output computation for each attention head $h_t^{(h)}$ using the KV cache can be expressed as:

$$h_t^{(h)} = \text{Softmax}\left(q_t^{(h)}\left(K_{t-1}^{(h)} \parallel k_t^{(h)}\right)^\top / \sqrt{m}\right)\left(V_{t-1}^{(h)} \parallel v_t^{(h)}\right), \tag{2}$$

where $q_t^{(h)}, k_t^{(h)}, v_t^{(h)}$ represent the query, key, and value vectors for the new token embedding $x_t$. Here, $\parallel$ denotes concatenation along the sequence length dimension.

### 2.2 SPARSE APPROXIMATION

Given a dictionary, our goal is to efficiently decompose and represent KV cache, *i.e.*, approximate a state $k \in \mathbb{R}^m$ as a linear combination of a few vectors (atoms) from an overcomplete dictionary $D \in \mathbb{R}^{m \times N}$. This reconstruction is given by $k = Dy$, where $y \in \mathbb{R}^N$ is the sparse representation vector such that $s = ||y||_0$. For implementation, $y$ only requires space proportional to $s$, not $N$.

We hypothesize that the KV cache, like other domains where sparse approximation is effective, contains inherent redundancy that can be leveraged for efficient compression. For instance, Figure 2 presents pairwise cosine similarity plots for keys generated during inference on a random subset of the WikiText dataset. Here, we observe that key states cluster in multiple different subspaces. Dictionary learning can take advantage of such redundancy, enabling KV vectors to be represented by a compact set of atoms with only a few active coefficients.

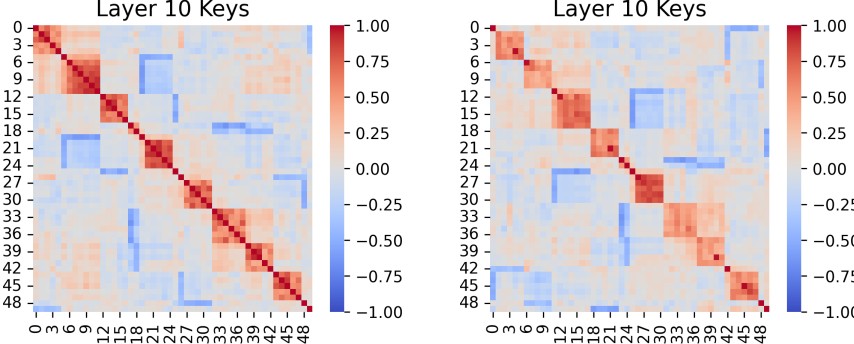

Figure 2: **Left** shows a pairwise cosine similarity matrix between key vectors generated from one input text from all heads in Layer 10 of Llama-3.1-8B-Instruct. Keys are sorted by similarity to demonstrate the clusters. **Right** shows the similarity matrix between key vectors from two *different* input texts. These plots indicate that there may exist a mixture of low-dimensional subspaces in the space of *all* possible keys, a hypothesis that naturally leads to dictionary learning.

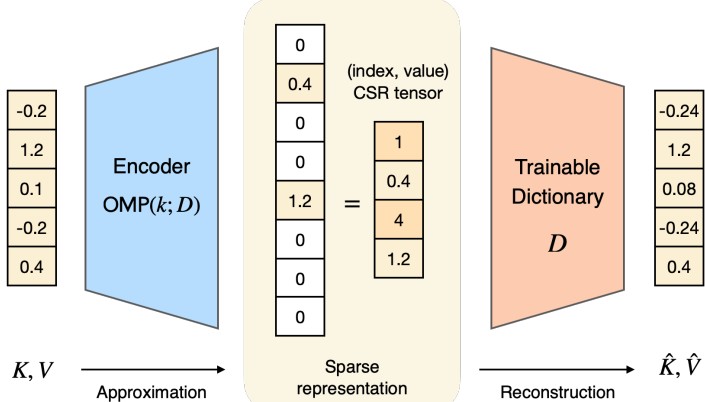

Figure 3: **Dictionary Learning.** We train a linear layer $D$ (our dictionary), that minimizes $\ell_2$-reconstruction error of KV states. The states of layer $i$ are used as training data for dictionary $D^{(i)}$. At each step, we apply OMP with fixed $D$ to represent KV as a vector of sparse coefficients; we then perform a step of gradient descent on $D$ and repeat the process. A sparse vector can be efficiently stored as a compressed sparse row (CSR), using a tuple of 16-bit index and 8-bit value.

Sparse approximation, which aims to find $y$ with minimum sparsity given $k$ and $D$, while ensuring a small reconstruction error, is NP-hard. This optimization problem is typically formulated as:

$$\min_{y} \|y\|_0 \text{ subject to } \|k - Dy\|_2 \leq \delta \|k\|_2 \text{ for some relative error threshold } \delta > 0 \qquad (3)$$

In this work, we adopt Orthogonal Matching Pursuit (OMP) as the sparse approximation algorithm. Given an input key or value vector $k$, a dictionary $D$, and a target sparsity $s$, OMP iteratively selects dictionary atoms to minimize the $\ell_2$-reconstruction error, with the process continuing until the specified sparsity $s$ is reached. Our implementation of OMP builds on advanced methods that utilize properties of the Cholesky inverse (Zhu et al., 2020) to optimize performance. Additionally, we incorporate implementation details from Lubonja et al. (2024) for efficient batched GPU execution and extend it to include an extra batch dimension, allowing for parallel processing across multiple dictionaries. The full algorithm is detailed in Appendix A.

## 2.3 Learning Layer-specific KV-Dict

**Layer-specific Dictionaries.** While the sparse approximation algorithm is crucial, achieving a high compression ratio relies heavily on well-constructed dictionaries. In this section, we describe the process for training the dictionaries used in KV-Dict. We adopt distinct dictionaries for the key and value vectors in each transformer layer due to their different functionalities. We denote the key and value dictionaries at each layer as $D_k$ and $D_v \in \mathbb{R}^{m \times N}$, where $N$ is the fixed dictionary size. With $N = 1024$, the dictionaries add an additional 16.8MB to the model's storage requirements for 7B/8B models.

As shown in Figure 3, we train layer-specific KV dictionaries through direct gradient-based optimization. For a given key or value vector, denoted as $k \in \mathbb{R}^m$ and a dictionary $D \in \mathbb{R}^{m \times N}$, the OMP algorithm approximates the sparse representation $y \in \mathbb{R}^N$. This process is parallelized across multiple dictionaries, but for simplicity, we present the notation for a single dictionary. The dictionary training objective minimizes the $\ell_2$ norm of the reconstruction error, with the loss function $\mathcal{L} = \|k - Dy\|_2^2$. We enforce unit norm constraints on the dictionary atoms by removing any gradient components parallel to the atoms before applying updates.

**Training.** The dictionaries are trained on KV pairs generated from the WikiText-103 dataset using Adam (Kingma & Ba, 2014) with a learning rate of 0.0001 and a cosine decay schedule over 20 epochs. The dictionaries are initialized with a uniform distribution, following the default initialization method for linear layers in PyTorch. For Llama-3.1-8B-Instruct, with a sparsity of $s = 32$ and a dictionary size of $N = 1024$, the training process takes about 2 hours on an NVIDIA A100 GPU.

We demonstrate our trained dictionaries reconstruct and generalize better than dictionaries trained using sparse autoencoders (similarly to those from Makhzani & Frey (2013); Bricken et al. (2023)) across several corpora in Table 1. Our method consistently achieves lower relative reconstruction errors, such as $0.19 \pm 0.05$ on out-of-domain dataset CNN/DailyMail, and this trend is consistent across other datasets.

Despite being trained only on WikiText-103, KV-Dict demonstrates a degree of universality: our dictionaries achieve lower test loss on out-of-domain datasets such as TweetEval than the test loss on WikiText-103 for sparse autoencoders, offering significant compression with minimal reconstruction error. In the next subsection, we explore how low $\ell_2$-reconstruction loss translates to strong performance preservation in language modeling.

## 2.4 PREFILLING AND DECODING WITH KV-DICT

During the prefilling stage, each layer generates the KV vectors for the input tokens, as illustrated in Figure 1a. KV-Dict uses full-precision KV vectors for attention computation, which are then passed to subsequent layers. Subsequently, OMP finds the sparse representations of the KV vectors using layer-specific key and value dictionaries, $\boldsymbol{D}_k$ and $\boldsymbol{D}_v$.

The compressed key and value caches are denoted as $\boldsymbol{K}_{\text{csr}}, \boldsymbol{V}_{\text{csr}} \in \mathbb{R}^{l_{\text{seq}} \times N}$ and replace the full-precision KV cache. The reconstruction of the KV cache is performed as follows:

$$\hat{\boldsymbol{K}} = \boldsymbol{K}_{\text{csr}} \boldsymbol{D}_k^\top, \quad \hat{\boldsymbol{V}} = \boldsymbol{V}_{\text{csr}} \boldsymbol{D}_v^\top \tag{4}$$

Recall that at the $t$-th iteration of autoregressive decoding, each layer receives $\boldsymbol{q}_t$, $\boldsymbol{k}_t$, and $\boldsymbol{v}_t$, the query, key, and value vectors corresponding to the newly generated token. Similarly to prior work (Liu et al., 2024b; Kang et al., 2024), we find that keeping a small number of recent tokens in full precision improves the generative performance of the model. To achieve this, we introduce a buffer that temporarily stores recent tokens in an uncompressed state. The KV vectors stored in the buffer are denoted as $\boldsymbol{K}_{\text{buffer}}, \boldsymbol{V}_{\text{buffer}} \in \mathbb{R}^{n_b \times m}$, where $n_b$ is the number of KV vectors in the buffer. The key cache up to, but not including the new token at iteration $t$, is then reconstructed as follows:

$$\hat{\boldsymbol{K}}_{t-1} = \boldsymbol{K}_{\text{csr}} \boldsymbol{D}_k^\top \parallel \boldsymbol{K}_{\text{buffer}} \tag{5}$$

Substituting this reconstruction into the Equation 2, the attention weights for each head $\boldsymbol{a}_t^{(h)}$ are computed as:

$$\boldsymbol{a}_t^{(h)} = \text{Softmax} \left( \boldsymbol{q}_t^{(h)} (\boldsymbol{K}_{\text{csr}}^{(h)} \boldsymbol{D}_k^\top \parallel \boldsymbol{K}_{\text{buffer}}^{(h)} \parallel \boldsymbol{k}_t^{(h)})^\top / \sqrt{m} \right) \tag{6}$$

A key implementation is that attention for the compressed sparse key cache and the uncompressed key cache is computed separately. For compressed sparse key cache, we first compute the product $\boldsymbol{q}_t^{(h)} \boldsymbol{D}_k$ before we multiply $\boldsymbol{K}_{\text{csr}}$, directly calculating the pre-softmax attention scores for compressed tokens. Attention for the buffer tokens is computed as usual. These scores are then concatenated with softmax to produce the final attention weights (Figure 1b). This process is formalized as:

$$\boldsymbol{a}_t^{(h)} = \text{Softmax} \left( \left( \boldsymbol{q}_t^{(h)} \boldsymbol{D}_k \boldsymbol{K}_{\text{csr}}^{(h)\top} \mid \boldsymbol{q}_t^{(h)} (\boldsymbol{K}_{\text{buffer}}^{(h)} \parallel \boldsymbol{k}_t^{(h)})^\top \right) / \sqrt{m} \right), \tag{7}$$

where $\mid$ represents concatenation along columns for attention scores.

Table 1: Relative reconstruction errors of different methods when training dictionaries of size 1024 and sparsity $s = 32$ on WikiText-103. Sparse Autoencoder is a two-layer perceptron with hard top-$k$ thresholding as activation (encoder as a linear layer + activation in Figure 3). KV-Dict is directly optimized using OMP as encoder. The KV states are generated from Llama-3.1-8B-Instruct model.

| Test Dataset | KV-Dict | Sparse Autoencoder | Random Dictionaries |
|---|---|---|---|
| WikiText-103 | $\mathbf{0.17} \pm 0.06$ | $0.20 \pm 0.05$ | $0.27 \pm 0.02$ |
| CNN/DailyMail | $\mathbf{0.19} \pm 0.05$ | $0.22 \pm 0.04$ | $0.27 \pm 0.02$ |
| IMDB | $\mathbf{0.18} \pm 0.05$ | $0.22 \pm 0.05$ | $0.27 \pm 0.02$ |
| TweetEval | $\mathbf{0.18} \pm 0.06$ | $0.21 \pm 0.05$ | $0.27 \pm 0.02$ |

When the buffer reaches capacity, OMP compresses the KV vectors for the oldest $n_a$ tokens in the buffer. This process is independent of the attention computation for the newest token and can therefore be performed in parallel.

**Time and Space Complexity.** The sparse representations are stored in CSR format, with values encoded in `FP8(E4M3)`, and all indices, including offsets, are stored as `int16`. Each row in CSR corresponds to a single key or value vector. For a given sparsity level $s$, the memory usage includes: nonzero values ($s$ bytes), dictionary indices ($2s$ bytes), and the offset array (2 bytes), resulting in a total size of $3s + 2$ bytes. For a head dimension of 128, a fully uncompressed vector using `FP16` takes 256 bytes, yielding a memory usage of $\frac{3s+2}{256} \times 100 \approx 1.17s\%$ (*e.g.*, 37.5% for $s = 32$).

In terms of time complexity, computing $\boldsymbol{q}_t \boldsymbol{K}_t^\top$ for a single head requires $O(l_{\text{seq}}m)$ multiplications. On the other hand, $\boldsymbol{q}_t \boldsymbol{D}_k \boldsymbol{K}_{\text{csr}}^\top$ needs $O(Nm + l_{\text{seq}}s)$ multiplications. This means that our computation is particularly well-suited for long-context tasks when $l_{\text{seq}} > m$ where $m$ is anywhere between 1024 and 4096. For short contexts when $l_{\text{seq}} < m$, our method only adds a small overhead to attention computation in actuality.

## 3 EXPERIMENTS

**Setup.** We evaluate our method on various models (Llama-3-8B, Llama-3.1-8B-Instruct, Llama-3.2-1B-Instruct, Llama-3.2-3B-Instruct, Mistral-7B-Instruct), using dictionaries trained on WikiText-103, as done in Section 2.3. To assess KV-Dict's effectiveness in memory reduction while maintaining long-context understanding, we conduct experiments on selected tasks from Long-Bench (Bai et al., 2023), following the setup of Liu et al. (2024b). See Appendix B for task details.

Additionally, we evaluate generative performance on complex reasoning tasks, such as GSM8K (Cobbe et al., 2021) with 5-shot prompting and MMLU-Pro Engineering/Law (Wang et al., 2024a) with zero-shot chain-of-thought. We compare our method against two kinds of KV cache compression methods: namely, quantization-based compression and eviction-based compression. For quantization-based methods, we evaluate KIVI (Liu et al., 2024b), ZipCache (He et al., 2024), and the Hugging Face implementation for per-token quantization. For eviction-based methods, we evaluate PyramidKV (Cai et al., 2024) and SnapKV (Li et al., 2024). We refer to the 4-bit and 2-bit versions of KIVI as KIVI-4 and KIVI-2, respectively, and denote its quantization group size as $g$.

We report KV size as the average percentage of the compressed cache relative to the full cache at the end of generation. KV-Dict's sparsity $s$ is set to match the KV size of the baseline.

**Hyperparameter Settings.** For both experiments, KV-Dict uses a dictionary size of $N = 4096$, a buffer size of $n_b = 128$, and an approximation window size $n_a = 1$, compressing the oldest token with each new token generated. For KIVI-4 and KIVI-2, we use a quantization group size of $g = 32$ and a buffer size of $n_b = 128$, as is tested and recommended in Liu et al. (2024b), for LongBench. For GSM8K and MMLU-Pro, we test for stronger memory savings, so we use $g = 64$ and $n_b = 64$ for KIVI.

### 3.1 EXPERIMENTAL RESULTS

**LongBench Results.** Table 2 presents the performance of KV-Dict and KIVI on LongBench tasks. KV-Dict demonstrates better performance than KIVI with similar or even smaller KV sizes. Notably, KV-Dict enables exploration of extremely low memory regimes that KIVI-2 cannot achieve. At a memory usage of just 12.4% KV size, KV-Dict maintains reasonable long-context understanding, with only 5.6%p and 4.4%p performance loss on Llama-3.1-8B-Instruct and on Mistral-7B-Instruct-v0.3, respectively, compared to the full cache (`FP16`). The largest performance loss comes from tasks with the lowest full cache accuracy, Qasper, yet there is almost no loss in simpler tasks, such as TriviaQA. This indicates that difficult tasks that require more complex understanding are much more sensitive to performance loss. Hence, it is important to evaluate on GSM8K, one of the harder natural language reasoning tasks, as we do next.

**GSM8K Results.** The performance of KV-Dict on GSM8K compared to KIVI is shown in Table 3.

Table 2: **Experimental results on LongBench.** For KV-Dict, we use $N = 4096$ as the dictionary size and $n_b = 128$ as the buffer size. For KIVI, we use $g = 32$ (group size for quantization) and $n_b = 128$ (buffer size). Sparsity level $s$ is set to match average KV size of KIVI, while $s = 8$ corresponds to cache size unattainable by common 2-bit quantizations. Full cache is in `FP16`.

| Method | KV Size | Qasper | QMSum | MultiNews | TREC | TriviaQA | SAMSum | LCC | RepoBench-P | Average |
|---|---|---|---|---|---|---|---|---|---|---|
| | | | | **Llama-3.1-8B-Instruct** | | | | | | |
| Full Cache | 100% | 22.54 | 24.57 | 27.44 | 72.5 | 91.65 | 43.47 | 63.15 | 56.76 | 50.26 |
| KIVI-4 | 33.2% | 22.83 | 23.72 | 27.95 | 71.0 | 90.39 | 44.25 | 62.93 | 55.48 | 49.78 |
| KV-Dict $_{s=24}$ | 30.6% | 21.68 | 24.25 | 27.20 | 72.5 | 91.58 | 42.93 | 62.92 | 56.51 | **49.95** |
| KIVI-2 | 21.1% | 13.77 | 22.72 | 27.35 | 71.0 | 90.85 | 43.53 | 62.03 | 53.00 | 48.03 |
| KV-Dict $_{s=16}$ | 21.4% | 15.45 | 23.13 | 25.78 | 72.5 | 92.25 | 42.02 | 63.01 | 55.58 | **48.71** |
| KV-Dict $_{s=8}$ | **12.4%** | 11.66 | 21.04 | 22.35 | 60.0 | 91.01 | 40.30 | 59.60 | 51.46 | 44.68 |
| | | | | **Mistral-7B-Instruct-v0.3** | | | | | | |
| Full Cache | 100% | 41.58 | 25.69 | 27.76 | 76.0 | 88.59 | 47.58 | 59.37 | 60.60 | 53.40 |
| KIVI-4 | 33.2% | 40.37 | 24.51 | 27.75 | 74.0 | 88.36 | 47.56 | 58.49 | 58.31 | 52.42 |
| KV-Dict $_{s=24}$ | 30.6% | 41.01 | 25.32 | 27.51 | 76.0 | 88.84 | 46.27 | 59.98 | 59.44 | **53.05** |
| KIVI-2 | 21.1% | 38.24 | 24.08 | 26.99 | 74.5 | 88.34 | 47.66 | 57.51 | 56.46 | 51.72 |
| KV-Dict $_{s=16}$ | 21.4% | 40.34 | 24.97 | 26.36 | 76.0 | 89.31 | 45.84 | 59.31 | 59.50 | **52.70** |
| KV-Dict $_{s=8}$ | **12.4%** | 33.03 | 22.80 | 22.85 | 68.5 | 87.85 | 43.10 | 56.66 | 56.85 | 48.96 |

Table 3: **Experimental results on GSM8K.** For KV-Dict, we use $N = 4096$ as the dictionary size and $n_b = 128$ as the buffer size. For KIVI, we use $g = 64$ (group size for quantization) and $n_b = 64$ (buffer size). Sparsity level $s$ is set to match the average KV size of KIVI, while $s = 4$ corresponds to cache size unattainable by common 2-bit quantizations. Full cache is in `FP16`.

(a) Llama 3.x 8B Models

| Method | KV Size | Llama-3-8B | 3.1-8B-Instruct |
|---|---|---|---|
| Full Cache | 100% | 49.89 | 79.61 |
| KIVI-4 | 38.2% | **49.13** | **78.17** |
| KV-Dict $_{s=24}$ | 36.9% | 48.29 | 76.88 |
| KIVI-2 | 25.7% | 40.56 | 67.93 |
| KV-Dict $_{s=14}$ | 26.1% | **48.75** | **75.06** |
| KV-Dict $_{s=4}$ | **15.8%** | 40.03 | 51.71 |

(b) Mistral 7B v0.3 Model

| Method | KV Size | 7B-Instruct |
|---|---|---|
| Full Cache | 100% | 48.60 |
| KIVI-4 | 38.2% | 48.52 |
| KV-Dict $_{s=20}$ | 32.7% | **48.60** |
| KIVI-2 | 25.7% | 42.91 |
| KV-Dict $_{s=10}$ | 22.0% | **44.35** |
| KV-Dict $_{s=4}$ | **15.8%** | 39.20 |

With a KV size of 36.9%, KV-Dict on Llama 8B models experiences a slight accuracy drop of less than $3\%p$, underperforming KIVI-4 at a similar KV size. However, in the lower memory regime near 25% KV size, KV-Dict significantly outperforms KIVI-2, achieving a higher accuracy by $8.2\%p$ on the Llama-3-8B model and $7.1\%p$ on the Llama-3.1-8B-Instruct model. These results highlight the robustness of KV-Dict in low-memory settings, demonstrating that low reconstruction error can be achieved using only a few atoms from our universal dictionary. To further test the resilience of KV-Dict, we set the sparsity to $s = 4$, observing a noticeable drop in accuracy on the Llama-3.1-8B-Instruct model. Despite this, both Llama models maintain an accuracy above 40%, which is remarkable given that only 4 atoms from KV-Dict were used for each key-value state, utilizing just 15.8% of the full cache, including the buffer.

The performance of KV-Dict on the Mistral-7B-Instruct model is even more impressive. We demonstrate that for Mistral, KV-Dict not only outperforms KIVI-4 and KIVI-2 but also achieves higher accuracy with even less memory usage. We also evaluate KV-Dict with $s = 4$ on the Mistral model and observe an accuracy of 39.2%, further demonstrating robustness in low-memory settings.

**Results across model sizes and baselines.** We illustrate the trade-off between memory usage and performance across six different KV cache compression methods on Llama models (1B, 3B, and 8B) in Figure 4. For all three model sizes, KV-Dict consistently lies on the Pareto frontier, achieving higher scores than other compression methods at similar KV cache budget sizes. Notably, KV-Dict demonstrates greater robustness at smaller model scales, with larger performance gaps observed

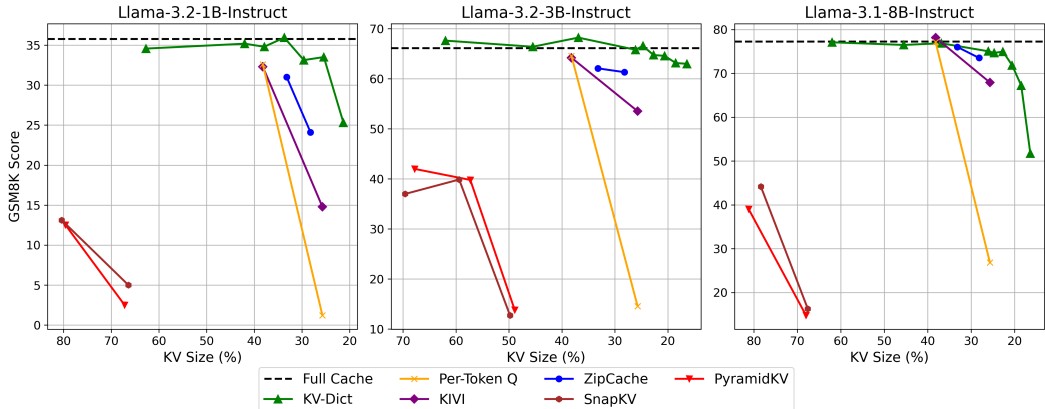

Figure 4: **Memory usage vs. performance of KV-Dict compared to other KV cache compression methods on GSM8K.** The figure illustrates the relationship between KV cache size and the performance of KV-Dict on Llama models on GSM8K. For KV-Dict, we use $N = 4096$ as the dictionary size and $n_b = 128$ as the buffer size. KV-Dict consistently outperforms both eviction-based methods (SnapKV, PyramidKV) and quantization-based methods (per-token quantization, KIVI, ZipCache).

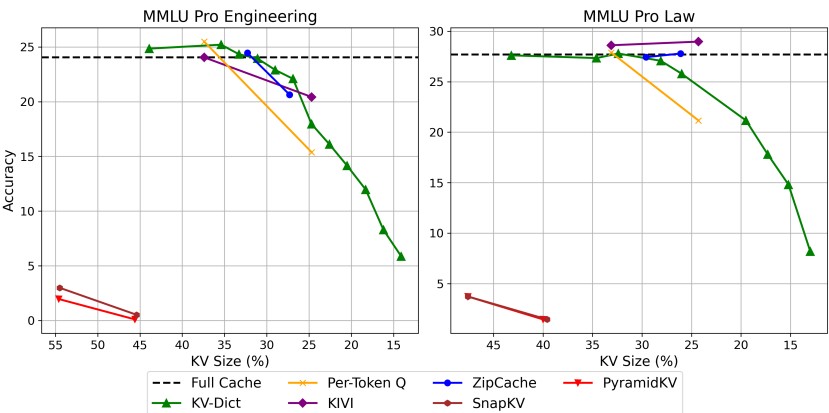

Figure 5: **Memory usage vs. performance of KV-Dict on MMLU-Pro.** The figure illustrates the relationship between KV cache size and the performance of KV-Dict on Llama models on MMLU-Pro Engineering/Law. For KV-Dict, we use $N = 4096$ as the dictionary size and $n_b = 128$ as the buffer size. KV-Dict often outperforms both eviction-based methods (SnapKV, PyramidKV) and quantization-based methods (per-token quantization, KIVI, ZipCache). For Law, our method slightly underperforms around 25%, but in lower memory regimes, our method still outperforms any other baseline.

for the 1B and 3B models. In the extremely low-memory regime below 20%, where quantization methods such as KIVI and ZipCache cannot achieve feasible cache sizes, KV-Dict achieves superior performance. Furthermore, while eviction-based methods (SnapKV, PyramidKV) can operate in these extremely low-memory settings, their performance lags significantly behind due to their incompatibility with GQA, making KV-Dict the effective choice for stringent memory constraints.

**MMLU-Pro Results.** Figure 5 illustrates the trade-offs between memory usage and performance for KV-Dict on the MMLU-Pro Engineering and Law subjects using the Llama-3.1-8B-Instruct model. KV-Dict outperforms eviction-based methods like SnapKV and PyramidKV across all memory settings, though its performance is comparable to quantization-based methods such as KIVI and ZipCache. In a low memory regime below 20% cache, however, our method still outperforms any other baseline. This highlights that KV-Dict supports a wide range of compression ratios quite effectively and that our dictionary is generalizable across input distribution.

## 3.2 ABLATION STUDY

**Error thresholding in sparse approximation.**    KV-Dict also supports a quality-controlled method for memory saving by allowing early termination of the sparse approximation process when a pre-defined error threshold is met. This approach conserves memory that would otherwise be used for marginal improvements in approximation quality. Detailed descriptions and results of this ablation study are provided in the Appendix C.1.

**Performance without buffer.**    To evaluate the impact of the buffer, we first conducted experiments with varying sparsity without the buffer, with the results shown by the dashed lines in Figure 6 in Appendix C.2. The comparison shows that removing the buffer results in a more pronounced decline in performance, especially at lower KV sizes.

**Balancing memory between buffer and sparse representation.**    Additionally, as shown in Table 4, we examine how balancing memory allocation between the buffer and the sparse representation affects performance. By fixing the total KV cache size at 25% of the original, we vary the memory distribution between the buffer and the sparse representation across three LongBench tasks: Qasper, MultiNews, and TREC. The results demonstrate that long-context understanding ability while using KV-Dict is not solely reliant on the buffer or the sparse representation. Rather, there exist optimal balance points where performance is maximized for each task.

Table 4: **Balancing memory between buffer and sparse representation.** This table shows the performance of KV-Dict with the Llama-3.1-8B-Instruct model on LongBench tasks (Qasper, Multi-News, TREC) while varying the memory allocation between the buffer and the sparse representation, with the total KV cache size fixed at 25% of the original size.

| Qasper | | | MultiNews | | | TREC | | |
|---|---|---|---|---|---|---|---|---|
| s | $n_b$ | F1 Score | s | $n_b$ | ROUGE-L | s | $n_b$ | Accuracy |
| 1 | 862 | 6.38 | 1 | 503 | 17.20 | 1 | 1232 | 58.5 |
| 4 | 724 | 8.36 | 4 | 423 | 20.21 | 4 | 1035 | 63.5 |
| 8 | 517 | 14.58 | 8 | 302 | 21.27 | **8** | **739** | **65.0** |
| **12** | **278** | **17.84** | **12** | **163** | **22.81** | 12 | 398 | 63.5 |
| 16 | 0 | 8.27 | 16 | 0 | 10.70 | 16 | 0 | 54.5 |

**Adaptive dictionary learning.**    While our universal dictionaries demonstrate strong performance, we explore an adaptive learning method to better incorporate input-specific context. This adaptive approach improves performance by adding new dictionary atoms during generation when the predefined reconstruction error threshold is not met. These atoms, tailored to the input prompt, improve performance but cannot be shared across batches, requiring them to be included in the KV size calculation. While this approach boosts accuracy, it increases memory usage, limiting its ability to achieve low-memory regimes. Detailed methods and results are provided in Appendix C.3.

## 3.3 LATENCY ANALYSIS

In this section, we present latency measurements of the forward pass and OMP portion of KV-Dict during decoding stage in Table 5. We run simple generation tests on a 1000 token input to Llama-3.1-8B-Instruct model and generate up to 250 tokens to measure and aggregate latency metrics. We compare both dictionary sizes $N = 1024$ and 4096, which primarily affects OMP computation time. We set the sparsity level to $s = 24$, and process OMP in batches of $n_a = 8$.

Although we list the forward pass and OMP separately, the processes are implemented to run in parallel such that the one generation step takes the maximum of the two durations plus some overhead. However, with parallelization, there exists a time versus space complexity tradeoff, since running OMP also consumes GPU memory.

Table 5: **Latency measurements.** The following latencies measure the total time it takes for the respective computation to process when generating one new token. We use Llama-3.1-8B-Instruct and sum up the time each operation takes in total across all 32 layers. Latencies when dictionary sizes are 1024 and 4096 are measured. Detailed settings are described in Section 3.3.

| Computation Type | Latency (per token) | |
| --- | --- | --- |
| - | $N = 1024$ | $N = 4096$ |
| Standard forward pass ($\boldsymbol{q}\boldsymbol{K}^\top$) | 48.39 ms | – |
| KV-Dict: forward pass using $\boldsymbol{q}(\boldsymbol{K}_{\mathrm{csr}}\boldsymbol{D}_k^\top)^\top$ | 55.56 ms | 56.35 ms |
| KV-Dict: sparse approximation via OMP | 26.57 ms | 40.58 ms |

## 4 RELATED WORK

Prior work on KV cache optimization has explored both training-stage and deployment-focused strategies to improve the efficiency of LLMs. On the deployment side, Kwon et al. (2023) introduce a Paged Attention mechanism and the popular vLLM framework, which adapt CPU-style page memory to map KV caches onto GPU memory using a mapping table, thereby minimizing memory fragmentation and leveraging custom CUDA kernels for efficient inference. While there is a significant and important line of research in this direction (Lin et al., 2024; Qin et al., 2024), this direction is orthogonal to our work and can often be used in tandem with quantization.

Current post-training KV cache compression methods can broadly be categorized into eviction, quantization, and merging. Zhang et al. (2024) introduced H2O, which uses attention scores to selectively retain tokens while preserving recent ones that are strongly correlated with current tokens. Multiple works discuss various heuristics and algorithms to find which tokens can be discarded, while some works find how to complement evictions methods (Ge et al., 2023; Li et al., 2024; Liu et al., 2024a; Devoto et al., 2024; Dong et al., 2024). For this line of work, there is a chance that evictions can work well together with KV-Dict, as Liu et al. (2024a) have successfully combined quantization and eviction.

Quantization methods have also played a crucial role in reducing KV cache size without compromising model performance. Although there is a flurry of work, we only mention those that are most relevant to our discussion and methodology. Hooper et al. (2024) identified outlier channels in key matrices and developed KVQuant, while Liu et al. (2024b) pursue a similar per-channel strategy in KIVI. Further extending these ideas, Yue et al. (2024) presented WKVQuant, which quantizes model weights as well as KV cache using two-dimensional quantization. Kang et al. (2024) follow similar per-channel key and per-token value quantization as KIVI, but with additional low-rank and sparse structures to manage quantization errors.

## 5 CONCLUSION

In conclusion, our proposed method, KV-Dict, offers a novel approach to compressing KV cache for transformers by leveraging low-dimensional structures and sparse dictionary learning. Through this method, we demonstrate that substantial redundancy exists among key states across various inputs, allowing us to compress the KV cache efficiently while maintaining near-lossless performance. Furthermore, KV-Dict enables compression rates that surpass traditional quantization techniques, offering fine-grained and wide control over memory usage. Importantly, our universal dictionary is both compact and scalable, making it applicable across tasks and user inputs without increasing memory demands. This approach provides strong memory savings, particularly for long-context tasks, by alleviating the memory bottlenecks associated with KV cache storage without dropping any previous tokens.

Future research directions based on our work include optimizing CSR tensors through customized quantizations and improving latency tradeoffs that occur due to the use of OMP during prefilling and decoding. It would also be interesting to apply "soft-eviction strategies" for sparse tensors in which sparsity level $s$ is determined or dropped later on based on the estimated importance of the token. A dynamic allocation of sparsity can further improve our compression method.

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

APPENDIX

## A  IMPLEMENTATION DETAILS

Algorithm 1 illustrates a naive implementation of OMP for understanding. In KV-Dict, we adopt the implementation of OMP v0 proposed by (Zhu et al., 2020), which minimizes computational complexity using efficient inverse Cholesky factorization. Additionally, we integrate methods from (Lubonja et al., 2024) for batched GPU execution and extend the implementation to handle multiple dictionaries in parallel.

---

**Algorithm 1** OMP

---

**Require:** Signal $\boldsymbol{k} \in \mathbb{R}^m$, dictionary $\boldsymbol{D} \in \mathbb{R}^{m \times N}$, sparsity $s$
**Ensure:** Sparse representation $\boldsymbol{y} \in \mathbb{R}^n$
1: Initialize $\boldsymbol{r}^{(0)} \leftarrow \boldsymbol{k}, \mathbb{I}^{(0)} \leftarrow \emptyset, \boldsymbol{y}^{(0)} \leftarrow \boldsymbol{0}$
2: **for** $i = 1$ to $s$ **do**
3: $\quad n^{(i)} \leftarrow \arg\max_{1 \leq n \leq N} \left\{ \left| \left( \boldsymbol{D}^\top \left( \boldsymbol{k} - \boldsymbol{D}\boldsymbol{y}^{(i)} \right) \right)_n \right| \right\}$
4: $\quad \mathbb{I}^{(i)} \leftarrow \mathbb{I}^{(i-1)} \cup \left\{ n^{(i)} \right\}$
5: $\quad \boldsymbol{y}^{(i+1)} \leftarrow \arg\min_{\boldsymbol{y} \in \mathbb{R}^N} \left\{ \|\boldsymbol{k} - \boldsymbol{D}\boldsymbol{y}\|_2, \mathrm{Supp}\left( \boldsymbol{y} \right) \subset \mathbb{I}^{(i)} \right\}$
6: **end for**
7: **return** $\boldsymbol{y}$

---

**Algorithm 2** Prefilling and decoding with KV-Dict

---

1: **Parameter:** sparsity $s$, buffer length $n_b$, approximation length $n_a$

2: **procedure** PREFILLING
3: $\quad$ **Input:** $\boldsymbol{X} \in \mathbb{R}^{l_{\mathrm{seq}} \times d}$
4: $\quad \boldsymbol{K} \leftarrow \boldsymbol{X}\boldsymbol{W}_k, \boldsymbol{V} \leftarrow \boldsymbol{X}\boldsymbol{W}_v$
5: $\quad \boldsymbol{K}_{\mathrm{csr}} \leftarrow \mathrm{OMP}\left( \boldsymbol{K} \left[ : l_{\mathrm{seq}} - n_b \right], \boldsymbol{D}_k, s \right)$
6: $\quad \boldsymbol{V}_{\mathrm{csr}} \leftarrow \mathrm{OMP}\left( \boldsymbol{V} \left[ : l_{\mathrm{seq}} - n_b \right], \boldsymbol{D}_v, s \right)$
7: $\quad \boldsymbol{K}_{\mathrm{buffer}} \leftarrow \boldsymbol{K} \left[ l_{\mathrm{seq}} - n_b : \right], \boldsymbol{V}_{\mathrm{buffer}} \leftarrow \boldsymbol{V} \left[ l_{\mathrm{seq}} - n_b : \right]$
8: $\quad$ KV cache $\leftarrow \boldsymbol{K}_{\mathrm{csr}}, \boldsymbol{K}_{\mathrm{buffer}}, \boldsymbol{V}_{\mathrm{csr}}, \boldsymbol{V}_{\mathrm{buffer}}$
9: $\quad$ **return** $\boldsymbol{K}, \boldsymbol{V}$
10: **end procedure**

11: **procedure** DECODING
12: $\quad$ **Input:** KV cache, $\boldsymbol{x}_t \in \mathbb{R}^{1 \times d}$
13: $\quad \boldsymbol{q}_t \leftarrow \boldsymbol{x}_t \boldsymbol{W}_q, \boldsymbol{k}_t \leftarrow \boldsymbol{x}_t \boldsymbol{W}_k, \boldsymbol{v}_t \leftarrow \boldsymbol{x}_t \boldsymbol{W}_v$
14: $\quad \boldsymbol{K}_{\mathrm{csr}}, \boldsymbol{K}_{\mathrm{buffer}}, \boldsymbol{V}_{\mathrm{csr}}, \boldsymbol{V}_{\mathrm{buffer}} \leftarrow$ KV cache
15: $\quad \boldsymbol{K}_{\mathrm{buffer}} \leftarrow \mathrm{Concat}\left( \left[ \boldsymbol{K}_{\mathrm{buffer}}, \boldsymbol{k}_t \right], \dim = \mathrm{token} \right)$
16: $\quad \boldsymbol{V}_{\mathrm{buffer}} \leftarrow \mathrm{Concat}\left( \left[ \boldsymbol{V}_{\mathrm{buffer}}, \boldsymbol{v}_t \right], \dim = \mathrm{token} \right)$
17: $\quad \boldsymbol{a}_t \leftarrow \mathrm{Concat}\left( \left[ \boldsymbol{q}_t \boldsymbol{D}_k \boldsymbol{K}_{\mathrm{csr}}, \boldsymbol{q}_t \boldsymbol{K}_{\mathrm{buffer}} \right], \dim = \mathrm{token} \right)$
18: $\quad \boldsymbol{a}_t \leftarrow \mathrm{Softmax}\left( \boldsymbol{a}_t \right)$
19: $\quad \boldsymbol{V} \leftarrow \mathrm{Concat}\left( \left[ \boldsymbol{D}_v \boldsymbol{V}_{\mathrm{csr}}, \boldsymbol{V}_{\mathrm{buffer}} \right], \dim = \mathrm{token} \right)$
20: $\quad \boldsymbol{o}_t \leftarrow \boldsymbol{a}_t \boldsymbol{V}$
21: $\quad$ **if** $\mathrm{len}\left( \boldsymbol{K}_{\mathrm{buffer}} \right) > n_b$ **then**
22: $\quad\quad \boldsymbol{K}'_{\mathrm{csr}} \leftarrow \mathrm{OMP}\left( \boldsymbol{K}_{\mathrm{buffer}} \left[ : n_a \right], \boldsymbol{D}_k, s \right)$
23: $\quad\quad \boldsymbol{V}'_{\mathrm{csr}} \leftarrow \mathrm{OMP}\left( \boldsymbol{V}_{\mathrm{buffer}} \left[ : n_a \right], \boldsymbol{D}_v, s \right)$
24: $\quad\quad \boldsymbol{K}_{\mathrm{csr}} \leftarrow \mathrm{Concat}\left( \left[ \boldsymbol{K}_{\mathrm{csr}}, \boldsymbol{K}'_{\mathrm{csr}} \right], \dim = \mathrm{token} \right)$
25: $\quad\quad \boldsymbol{V}_{\mathrm{csr}} \leftarrow \mathrm{Concat}\left( \left[ \boldsymbol{V}_{\mathrm{csr}}, \boldsymbol{V}'_{\mathrm{csr}} \right], \dim = \mathrm{token} \right)$
26: $\quad\quad \boldsymbol{K}_{\mathrm{buffer}} \leftarrow \boldsymbol{K}_{\mathrm{buffer}} \left[ n_a : \right], \boldsymbol{V}_{\mathrm{buffer}} \leftarrow \boldsymbol{V}_{\mathrm{buffer}} \left[ n_a : \right]$
27: $\quad$ **end if**
28: $\quad$ KV cache $\leftarrow \boldsymbol{K}_{\mathrm{csr}}, \boldsymbol{K}_{\mathrm{buffer}}, \boldsymbol{V}_{\mathrm{csr}}, \boldsymbol{V}_{\mathrm{buffer}}$
29: $\quad$ **return** $\boldsymbol{o}_t$
30: **end procedure**

---

# B  LONGBENCH TASK STATISTICS

Table 6: **Details of LongBench tasks used in experiments.**

| Task | Task Type | Evaluation Metric | Average Length | # of Samples |
|------|-----------|-------------------|----------------|--------------|
| Qasper | Single-doc QA | F1 | 3619 | 200 |
| QMSum | Summarization | ROUGE-L | 10614 | 200 |
| MultiNews | Summarization | ROUGE-L | 2113 | 200 |
| TREC | Few-shot information retrieval | Accuracy | 5177 | 200 |
| TriviaQA | Few-shot reading comprehension | F1 | 8209 | 200 |
| SAMSum | Few-shot dialogue summarization | ROUGE-L | 6258 | 200 |
| LCC | Code completion | Edit Similarity | 1235 | 500 |
| RepoBench-P | Code completion | Edit Similarity | 4206 | 500 |

# C  ABLATION STUDY: EXPERIMENTAL RESULTS

## C.1  ERROR THRESHOLDING IN SPARSE APPROXIMATION

For the error thresholding ablation study, detailed results are provided in Table 7. We set a maximum sparsity of 32, corresponding to the maximum number of iterations for the OMP algorithm. However, if the reconstruction error at any iteration falls below a predefined error threshold, we let the OMP terminate early, saving memory that would otherwise be used for minor approximation improvements. This approach is particularly compatible with OMP, as its greedy nature ensures that early termination yields the same results as using higher sparsity (less non-zero elements). Additionally, OMP inherently computes the residual at each iteration, allowing for continuous evaluation of the relative reconstruction error without requiring any additional computation.

Table 7: **Impact of error thresholding on LongBench performance and memory usage.** The table presents the performance of KV-Dict on the Llama-3.1-8B-Instruct model at various reconstruction error thresholds ($\delta$) for early termination of the sparse approximation algorithm. A dictionary size of $N = 1024$ and `FP16` precision for the values of the CSR tensors are used.

| Threshold ($\delta$) | KV Size | Qasper | QMSum | MultiNews | TREC | TriviaQA | SAMSum | LCC | RepoBench-P | Average |
|----------------------|---------|--------|-------|-----------|------|----------|--------|-----|-------------|---------|
| | | | | **Llama-3.1-8B-Instruct** | | | | | | |
| Full Cache | 100% | 22.54 | 24.57 | 27.44 | 72.5 | 91.65 | 43.47 | 63.15 | 56.76 | 50.26 |
| 0.2 | 50.6% | 20.03 | 23.65 | 26.44 | 72.5 | 91.61 | 43.47 | 62.72 | 56.63 | 49.63 |
| 0.3 | 41.1% | 16.49 | 23.35 | 25.34 | 72.5 | 91.34 | 43.02 | 62.53 | 56.65 | 48.90 |
| 0.4 | 30.9% | 16.08 | 22.91 | 23.77 | 69.5 | 90.79 | 42.70 | 61.28 | 54.82 | 47.73 |
| 0.5 | 22.8% | 12.43 | 21.75 | 21.29 | 57.5 | 88.56 | 41.04 | 58.85 | 53.19 | 44.33 |

## C.2  PERFORMANCE WITHOUT BUFFER

In this section, we assess the effect of the buffer by comparing results with and without its use. The results for LongBench and GSM8K are presented in Table 8 and Table 9, respectively.

## C.3  ADAPTIVE KV-DICT

While we observe some degree of universality in our dictionaries, as shown in Table 1, their performance is particularly strong on WikiText-103, the dataset they were trained on. To better incorporate input context information, we propose an extension that adaptively learns the dictionary during generation. In this framework, we begin with a pre-trained universal dictionary as the initial dictionary. If, during the generation process, the sparse approximation fails to meet the predefined relative reconstruction error threshold, the problematic uncompressed key or value vector is normalized and added to the dictionary. The sparse representation of this vector is then stored with a sparsity of $s = 1$, where its index corresponds to the newly added atom and its value is the $\ell_2$ norm of the uncompressed vector. The updated dictionary is subsequently used for further sparse approximations during the generation task. In this way, the adaptive learning framework incrementally refines the

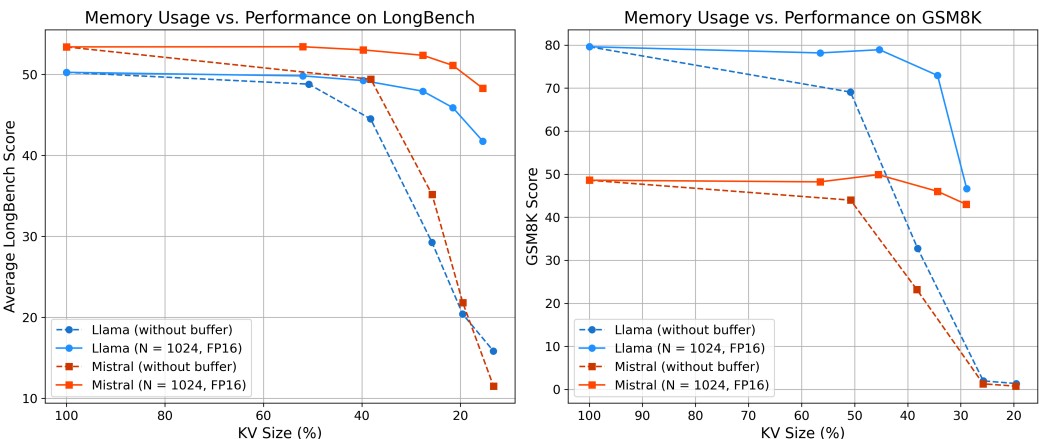

Figure 6: **Memory usage vs. performance of KV-Dict with and without buffer on LongBench and GSM8K.** The figure illustrates the impact of removing the buffer on the performance of KV-Dict when evaluated on the Llama 3.1-8B-Instruct and Mistral-7B-Instruct models for LongBench (left) and GSM8K (right) tasks. Solid lines represent configurations with a buffer, while dashed lines represent configurations without a buffer. We use a dictionary size of $N = 1024$ and `FP16` precision for the values of CSR tensors to vary sparsity and explore a wide range of KV sizes.

Table 8: **LongBench performance without buffer.** This table shows the impact of removing the buffer of KV-Dict on the performance of the Llama-3.1-8B-Instruct and Mistral-7B-Instruct models at varying sparsity levels. A dictionary size of $N = 1024$ and `FP16` precision for the values of the CSR tensors are used.

| Sparsity | KV Size | Qasper | QMSum | MultiNews | TREC | TriviaQA | SAMSum | LCC | RepoBench-P | Average |
|---|---|---|---|---|---|---|---|---|---|---|
| | | | | **Llama-3.1-8B-Instruct** | | | | | | |
| Full Cache | 100% | 13.10 | 23.46 | 26.94 | 72.5 | 91.65 | 43.47 | 63.15 | 56.76 | 48.88 |
| $s = 32$ | 50.8% | 14.87 | 26.51 | 26.57 | 71.5 | 92.48 | 42.88 | 61.54 | 54.04 | 48.80 |
| $s = 24$ | 38.2% | 13.37 | 25.02 | 22.54 | 65.0 | 91.75 | 39.71 | 52.21 | 46.48 | 44.51 |
| $s = 16$ | 25.8% | 8.27 | 13.74 | 10.70 | 54.5 | 77.51 | 20.45 | 26.53 | 22.46 | 29.27 |
| $s = 12$ | 19.5% | 6.31 | 10.15 | 5.66 | 39.0 | 53.70 | 6.83 | 22.18 | 19.46 | 20.41 |
| $s = 8$ | 13.3% | 2.74 | 8.05 | 4.17 | 36.5 | 34.45 | 4.27 | 18.24 | 18.32 | 15.84 |
| | | | | **Mistral-7B-Instruct-v0.3** | | | | | | |
| Full Cache | 100% | 41.58 | 25.69 | 27.76 | 76.0 | 88.59 | 47.58 | 59.37 | 60.60 | 53.40 |
| $s = 32$ | 50.8% | 40.27 | 25.21 | 27.53 | 76.5 | 89.01 | 45.77 | 58.64 | 59.07 | 52.75 |
| $s = 24$ | 38.2% | 37.46 | 24.41 | 27.34 | 75.5 | 88.66 | 43.87 | 48.55 | 49.50 | 49.41 |
| $s = 16$ | 25.8% | 25.57 | 18.49 | 15.19 | 71.5 | 81.91 | 27.90 | 19.39 | 21.45 | 35.18 |
| $s = 12$ | 19.5% | 18.59 | 13.11 | 5.95 | 58.0 | 50.13 | 2.86 | 13.38 | 12.60 | 21.83 |
| $s = 8$ | 13.3% | 10.32 | 6.98 | 2.67 | 31.5 | 20.01 | 2.27 | 10.18 | 8.11 | 11.51 |

Table 9: **GSM8K performance without buffer.** This table shows the impact of removing the buffer of KV-Dict on the performance of the Llama-3.1-8B-Instruct and Mistral-7B-Instruct models at varying sparsity levels. A dictionary size of $N = 1024$ and `FP16` precision for the values of the CSR tensors are used.

| Sparsity | KV Size | Llama-3.1-8B-Instruct | Mistral-7B-Instruct-v0.3 |
|---|---|---|---|
| Full Cache | 100% | 79.61 | 48.60 |
| $s = 32$ | 50.8% | 69.07 | 43.97 |
| $s = 24$ | 38.2% | 32.75 | 23.20 |
| $s = 16$ | 25.8% | 1.97 | 1.29 |
| $s = 12$ | 19.6% | 1.36 | 0.76 |

dictionaries, tailoring them to the specific generative task and enhancing overall performance at the cost of additional memory usage.

In our experiment, we use a universal dictionary of size 1024, allowing up to 1024 additional atoms to be added during generation. The maximum sparsity of 16 is used, with a buffer size of $n_b = 128$, and `FP16` precision for the values of the CSR tensors. Results of this experiment are presented in

Table 10: **GSM8K performance of adaptive KV-Dict.** The table shows the GSM8K performance and KV cache sizes of adaptive KV-Dict on the Llama-3.1-8B-Instruct and Mistral-7B-Instruct-v0.3 models at varying reconstruction error thresholds ($\delta$). A universal dictionary of size 1024 is used, with up to 1024 additional atoms added during generation. The maximum sparsity of $s = 16$, buffer size of $n_b = 128$, and `FP16` precision for CSR tensor values are applied.

| Threshold ($\delta$) | Llama-3.1-8B-Instruct | | Mistral-7B-Instruct-v0.3 | |
|---|---|---|---|---|
| | KV Size | GSM8K Score | KV Size | GSM8K Score |
| Full Cache | 100% | 79.61 | 100% | 48.60 |
| 0.25 | N/A | N/A | 42.1% | 48.07 |
| 0.30 | 43.5% | 77.41 | 41.3% | 48.14 |
| 0.35 | 42.0% | 76.80 | 39.8% | 47.76 |

# D  QUALITATIVE RESULTS

## D.1  GSM8K EXAMPLE GENERATION

The boxes highlight the correct intermeidate answers that must be derived. The highlighted colors in red show incorrect reasoning and answers. We provide the original problem and solution below, along with outputs from Llama-3.1-8B-Instruct with full KV cache (FP16), KV-Dict (sparsity $s = 14$), and KIVI-2.

---

**Original Problem & Solution**

**Problem:** The Kennel house keeps 3 German Shepherds and 2 Bulldogs. If a German Shepherd consumes 5 kilograms of dog food and a bulldog consumes 3 kilograms of dog food per day, how many kilograms of dog food will they need in a week?

**Solution:** The German Shepherds consume $3 \times 5 = \boxed{15}$ kilograms of dog food per day.

The Bulldogs consume $2 \times 3 = \boxed{6}$ kilograms of dog food per day.

The dogs consume a total of $15 + 6 = \boxed{21}$ kilograms of dog food per day.

Therefore, they need $21 \times 7 = \boxed{147}$ kilograms of dog food in a week.

**Final Answer:** 147

---

**Llama-3.1-8B-Instruct with Full Cache**

First, find the total amount of food the German Shepherds consume in a day: 3 dogs $\times$ 5 kg/dog = 15 kg.
Then find the total amount of food the Bulldogs consume in a day: 2 dogs $\times$ 3 kg/dog = 6 kg.
Add the two amounts together to find the total amount of food the dogs consume in a day: 15 kg $+6$ kg $= 21$ kg.
Multiply that number by 7 to find the total amount of food the dogs consume in a week: 21 kg/day $\times 7$ days $= 147$ kg.

---

**Llama-3.1-8B-Instruct with KV-Dict ($s = 14$)**

---

The kennel has 3 German Shepherds and 2 Bulldogs, so they have $3 + 2 = 5$ dogs.
The German Shepherds consume $5 \times 3 = 15$ kilograms of dog food per day.
The Bulldogs consume $2 \times 3 = 6$ kilograms of dog food per day.
So, the kennel needs $15 + 6 = 21$ kilograms of dog food per day.
In a week, they will need $21 \times 7 = 147$ kilograms of dog food.
The answer is 147.

**Llama-3.1-8B-Instruct with KIVI-2**

The Kennel house has 3 German Shepherds and 2 Bulldogs, so they have a total of $3 + 2 = \boxed{5}$ dogs.

Each German Shepherd consumes 5 kilograms of dog food per day, so the total amount of dog food consumed by the German Shepherds is $5 \times 5 = \boxed{25}$ kilograms per day.

Each Bulldog consumes 3 kilograms of dog food per day, so the total amount of dog food consumed by the Bulldogs is $2 \times 3 = \boxed{6}$ kilograms per day.

The total amount of dog food consumed per day is $25 + 6 = \boxed{31}$ kilograms.

The Kennel house will need $31 \times 7 = \boxed{217}$ kilograms of dog food in a week.

