# OpenReview forum: "KV-Dict: Sparse KV Cache Compression with Universal Dictionaries"
_ICLR.cc/2025/Conference — Submitted to ICLR 2025_

### Official Review · Reviewer_XfrT · 2024-10-31

**Soundness:** 2
**Presentation:** 2
**Contribution:** 3
**Rating:** 5
**Confidence:** 3

**Summary:**

This paper addresses the memory challenges of deploying Transformers in Large Language Models (LLMs), specifically focusing on the key-value (KV) cache, which grows with both model size and input length, consuming extensive GPU memory during inference. To tackle this, the authors introduce KV-Dict, a novel post-training compression technique for the KV cache. KV-Dict uses sparse dictionary learning to decompose and reconstruct key-value states, distinguishing itself from traditional quantization methods by providing flexible memory usage and fine-grained sparsity control with minimal performance loss.

**Strengths:**

1. This method uses a dictionary method for compression of KVCache, which is quite a less developed method in this topic

**Weaknesses:**

1. The problems stated in the introduction section are not well answered in the following sections. The paper stated two problems: “off the shelf” consideration and the problem of eviction strategies under long contexts. However, the experiments use KIVI, which is “off the shelf” (even more it doesn’t need training a dictionary) neither does it use eviction strategies.
2. The motivation is unclear. The introduction talked about the long context, but the abstract talked about reasoning tasks like GSM8k. Most questions+answers in GSM8k are below 800 characters, do we really need KVCache compression? how much gain can we get?
3. Generalizability issues: Based on Figure 2, it seems like 48 tokens contain 8 independent clusters, which requires at least 8 atoms. Then for N=4096, does it mean maybe it can only represent KVCache that are at most 24K long? Thus is this really Universal?
4. Value Cache issue: Still Figure 2, it actually gives a stronger implication than the author’s statements: the key cache is always similar to the key cache close to it, no matter what the content is. This is very strong, and does this also apply to the Value cache? If yes, this is counterintuitive, can you explain it? if not, is this method universal?

**Questions:**

The above questions as well as the following

1. For Table “top-k perceptron” what is “top” defined as? how do you select to make the autoencoder sparse? Since this seems weird that if the autoencoder converges why it cannot beat OMP which is a greedy method?

2. what about the Latency Comparison with KIVI?

3. For training, how do you initialize the dictionary? isn’t this important?
4. also for training, you didn’t specify clearly in the experiments section what is the training data for the dictionary, is it also wikitext as in Table 1? this is important to claim the generalizability of your method.

---

> ### Author Response · Authors · 2024-11-20
> **Authors' response to Reviewer XfrT (R4; part 1)**
>
> We thank the reviewer for recognizing the novelty of our work in exploring the use of dictionary learning in KV cache compression. We address each concern below.
>
> ### W1: Presentation regarding motivation and baseline methods
> We respectfully disagree with the reviewer’s concern that the problem highlighted in our introduction is not well-answered. As the reviewer noted, we emphasize that prior methods are not off-the-shelf and often discard the majority of tokens. Our method, KV-Dict, directly addresses these issues by being readily usable off-the-shelf and retaining all tokens without any discarding.
>
> To clarify any confusion, our choice of KIVI as a baseline may have led to misunderstanding, as KIVI already partially addresses these limitations. To resolve this, we have made two updates:
> - Revised introduction (L42-48): We explicitly outline the limitations of both eviction-based and quantization-based methods. Specifically, eviction methods can degrade long-context accuracy, while quantization methods have strict upper bounds on compression rates.
> - Expanded baselines: Our experiments now include multiple additional baselines, including PyramidKV and SnapKV for eviction-based methods, and ZipCache and per-token quantization for quantization-based methods.
>
>
> ### W2: Using GSM8K as a KV cache compression benchmark
> We respectfully disagree with the notion that GSM8K is not appropriate for measuring KV cache compression. In terms of its use as a benchmark, GSM8K is widely recognized in the literature as a standard benchmark for KV cache compression (e.g., [3,4,5,7]), particularly because reasoning and long-generation tasks are highly sensitive to even minimal degradation caused by compression. We also evaluate GSM8K with 5-shot prompting which is longer than just one GSM8K question. Additionally, we complement GSM8K with LongBench, which evaluates long-context tasks such as summarization, information retrieval, and document QA. These benchmarks collectively capture diverse dimensions of KV cache compression performance—GSM8K tests reasoning robustness during generation, while LongBench evaluates token preservation for long-context understanding. Together, they provide a comprehensive assessment necessary for modern LLMs.
>
>
>
>
> ### W3&4: Generalizability and value cache issue
> We believe the reviewer's concern stems from a misunderstanding of our setting. The elements in Figure 2 have been sorted by similarity to highlight the clustering behavior, as stated in L159-160. This sorting is not reflective of sequence order but rather aims to visualize the inherent redundancy and clustering among the key and value states.
>
> Regarding the generalizability concerns (W3), the clustering behavior observed does not imply that the number of clusters will scale proportionally with the sequence length. Instead, it highlights that key and value states naturally form clusters, irrespective of their positions within the sequence. This property is independent of sequence length and supports the universal nature of the method.
>
> Regarding the value cache concerns (W4), our statement is that both the key and value states exhibit clustering behavior, irrespective of their sequence positions, rather than being similar solely to neighboring states in the sequence. While the clustering effect is more pronounced in key states, a similar behavior is also evident in value states. This finding supports the universality of our method, though we acknowledge that clustering in value states is not as strong as in key states.

---

> > ### Author Response · Authors · 2024-11-20
> > **Authors' response to Reviewer XfrT (R4; part 2)**
> >
> > ### Q1: Dictionary learning and OMP
> > **Definition of "top-k" and its role in sparsity.**
> > In our experiments on training dictionaries, we utilize "top-k" hard thresholding as the activation function within the two MLP layers. This approach enforces sparsity by retaining only the top $k$ elements with the largest absolute values from the hidden states of length $N$, while setting the rest to zero. Formally, for $x \in \mathrm{R}^N$, $\text{topk}(x)$ zeros out all entries that are not top-k in magnitude, *e.g.*, $\text{topk}([4, -1, 10, -9]) = [0, 0, 10, -9]$ when $k=2$.
> >
> > We specifically chose top-k hard thresholding over alternatives like ReLU or L1 regularization because it directly aligns with our deployment objective: reconstructing KV states using a linear combination of a small number of dictionary atoms. This explicit control over sparsity makes top-k well-suited for the constraints of our task.
> >
> > **Why OMP outperforms the sparse autoencoder.**
> > Sparse approximation is a highly non-convex problem, and convergence alone does not guarantee optimality. The sparse autoencoder, with its limited two-layer architecture, can be viewed as simple approximation of dictionary learning [1], which reduces its capacity to effectively navigate the complex loss landscape. This limitation increases the likelihood of convergence to suboptimal solutions, which may fail to match the performance of more specialized algorithms.
> >
> > In contrast, OMP, despite its simplicity as a greedy algorithm, is backed by strong theoretical guarantees and is widely regarded as highly effective for sparse approximation tasks [2, 6]. We believe its robustness and suitability for this domain, while sparse autoencoders may converge to a higher loss regime.
> >
> >
> > ### Q2: Latency comparison with KIVI
> > We provide end-to-end latency measurements for prefilling and decoding as follows: with an input length of 1000 tokens and a batch size of 1, the standard implementation takes 0.14 seconds to prefill and 48.39 ms/token for decoding, while KIVI takes 0.15 seconds to prefill and 59.66 ms/token for decoding. Under the same settings, KV-Dict takes 6.98 seconds to prefill and 96.93 ms/token for decoding. It is important to note that the current implementation of our algorithm was not optimized for latency, and the majority of the observed latency stems from the use of a naive OMP. We believe that adopting more advanced approximation algorithms or leveraging hardware optimizations—similar to those employed by KIVI—has the potential to significantly improve latency.
> >
> > While we acknowledge that latency may be a concern for certain practitioners, it is important to underscore that the primary focus of our method is addressing highly memory-constrained scenarios. Such scenarios are increasingly critical for real-world LLM deployments, where even a batch size of one can exceed the capacity of a single GPU. By prioritizing memory efficiency, KV-Dict enables feasible deployment in contexts where other methods may encounter out-of-memory errors, offering a crucial contribution to memory-limited settings.
> >
> >
> > ### Q3&4: Initialization and training of dictionaries
> > Please refer to our General Response, where we address this question in our response to "Q1: How to initialize and train KV-Dict". For initialization, we simply initialized the dictionary just as we would do for a linear layer. While experimenting with different initialization schemes is a promising direction, we found our current initialization sufficient for the strong results we have obtained.
> >
> >
> > ### References
> >
> > [1] Bricken, T., Templeton, A., Batson, J., Chen, B., Jermyn, A., Conerly, T., Turner, N., Anil, C., Denison, C., Askell, A. and Lasenby, R., 2023. Towards monosemanticity: Decomposing language models with dictionary learning. Transformer Circuits Thread, 2.
> >
> > [2] Candes, E.J. and Tao, T., 2006. Near-optimal signal recovery from random projections: Universal encoding strategies?. IEEE transactions on information theory.
> >
> > [3] Ge, S., Zhang, Y., Liu, L., Zhang, M., Han, J. and Gao, J., Model Tells You What to Discard: Adaptive KV Cache Compression for LLMs. In The Twelfth International Conference on Learning Representations.
> >
> > [4] He, Y., Zhang, L., Wu, W., Liu, J., Zhou, H. and Zhuang, B., 2024. ZipCache: Accurate and Efficient KV Cache Quantization with Salient Token Identification. arXiv preprint arXiv:2405.14256.
> >
> > [5] Liu, Z., Yuan, J., Jin, H., Zhong, S., Xu, Z., Braverman, V., Chen, B. and Hu, X., KIVI: A Tuning-Free Asymmetric 2bit Quantization for KV Cache. In Forty-first International Conference on Machine Learning.
> >
> > [6] Tropp, J.A. and Gilbert, A.C., 2007. Signal recovery from random measurements via orthogonal matching pursuit. IEEE Transactions on information theory.
> >
> > [7] Yang, J.Y., Kim, B., Bae, J., Kwon, B., Park, G., Yang, E., Kwon, S.J. and Lee, D., 2024. No token left behind: Reliable kv cache compression via importance-aware mixed precision quantization. arXiv preprint arXiv:2402.18096.

---

> > ### Comment · Reviewer_XfrT · 2024-11-26
> >
> > Thanks for the reply. For W3, I just noticed you rank by similarity but not position. So you intend to justify "Can we find substantial redundancy among keys and values, universal of all inputs?"
> >
> >
> > Overall, you propose to map KV cache to another space and claim that they can be sparsely represented in that space.
> > suppose you are using s=16 and head dim is 128, that is 1/8 of compression which is similar to 2/16 bits for KIVI. And in this case, you get about 1% improvements on LongBench.
> >
> >
> > I feel like the improvement is quite weak. And the performance is degraded significantly under s=8.
> >
> > You have a higher compression ratio that is somewhat equivalent to 1-bit compression. This is the highlight part for me: you make the compression ratio further. It is better if you focus on this when writing the introduction. On the other hand, "universality" is too strong and actually it is not a sound contribution. (Quantization methods don't have this universality problem).
> >
> >
> > Now I am most concerned about the latency:
> > 1. seems like we need to add both the encoding and reconstruction parts in Table 5. which nearly doubles the latency.
> > 2. isn't 1000 tokens too short? How will it vary when it gets longer?
> >
> > If you want to improve the throughput by utilizing the saved memory, please perform that experiment instead of measuring latency.
> >
> > Overall the performance and latency concerns make it hard for me to raise the score.
> >
> > You can just show a few experiment results on the throughput improvements (best compared with KIVI) to save your time, if you still want to convince reviewers (at least for me). And I recommend you improve your paper as I stated above in the future.

---

> > > ### Author Response · Authors · 2024-11-28
> > > **Authors' response to Reviewer XfrT (part 1)**
> > >
> > > Thank you once again for your detailed reply. We appreciate the time and effort.
> > >
> > > First, we would like to confirm if the responses we have provided during the rebuttal phase adequately address the weaknesses you initially listed in your review; we endeavored to address each concern thoroughly and thoughtfully. If there are any areas where you feel our clarifications were insufficient or further elaboration would help, we would be more than happy to provide additional details.
> > >
> > > We understand that initial scores often reflect a preliminary assessment based on the identified weaknesses. Given the responses and the steps we’ve taken to address these points, we would like to know if this alleviates your concerns and justifies a score re-evaluation. We provide a response to performance and latency concerns below.
> > >
> > > ### Performance of KV-Dict
> > > > And in this case, you get about 1% improvements on LongBench. I feel like the improvement is quite weak.
> > >
> > > We would like to make sure that the reviewer has seen our additional experiments that have been included in the revised version. KV-Dict outperforms multiple state-of-the-art KV cache compression methods (page 8). We disagree that a 1%p improvement over KIVI on LongBench is weak, as the *full-cache performance is only 1.68%p higher than that of KIVI*.
> > > Since both methods exhibit near-lossless performance on LongBench, we wish to emphasize KV-Dict's significant performance gains on complex reasoning tasks, such as GSM8K.
> > > In these experiments, KV-Dict demonstrates substantial performance improvements across various model sizes (refer to Figure 4 in the revised version for the memory vs. performance curves). Specifically, we observed approximately 19%p, 12%p, and 5%p improvements on GSM8K compared to KIVI under similar KV cache budgets. Comparisons with other KV cache compression methods are also provided below:
> > >
> > > **GSM8K Results**
> > > | Model | Full cache (fp16) | KV-Dict | ZipCache | KIVI | Per-Token Q | PyramidKV | SnapKV |
> > > |-------|------------|---------|----------|------|-------------|-----------|--------|
> > > | Llama-3.2-1B-Instruct | 35.78 (100%) | **33.51** (25.4%) | 24.11 (28.2%) | 14.78 (25.7%) |  1.21 (25.7%) |  2.50 (67.2%)  |  5.00 (66.3%)  |
> > > | Llama-3.2-3B-Instruct | 66.11 (100%) | **65.81** (26.1%) | 61.33 (28.2%) | 53.53 (25.7%) | 14.56 (25.7%) | 13.80 (49.8%) | 12.74 (48.9%) |
> > > | Llama-3.1-8B-Instruct | 77.26 (100%) | **75.06** (26.1%) | 73.54 (28.2%) | 69.60 (25.7%) | 28.35 (25.7%) | 14.78 (68.0%) | 16.30 (67.4%) |
> > >
> > >
> > > ### Universality as an empirical finding and application
> > > The reviewer is correct that universality is not a concern for quantization methods. However, we would like to emphasize the significance of the universal dictionary in two key aspects:
> > >
> > > 1. **Empirical Finding:** We demonstrate that keys and values lie close to a union of subspaces regardless of the input text. We believe this observation of substantial redundancy in KV states across all inputs is a meaningful contribution in itself.
> > >
> > > 2. **Application:** To our knowledge, our work is the first to exploit this property by using a universal dictionary rather than constructing distinct low-rank subspaces for each key or value, as done in previous works. This approach underscores the novelty and potential impact of our method.

---

> > > > ### Author Response · Authors · 2024-11-28
> > > > **Authors' response to Reviewer XfrT (part 2)**
> > > >
> > > > ### Concerns on latency
> > > >
> > > > We appreciate the reviewer's feedback, particularly recognizing the key strength of our method: achieving compression rates beyond 2-bits, which represents a critical advancement for memory-constrained scenarios. The primary motivation behind KV-Dict is to address the increasing challenge posed by memory limitations in deploying large language models (LLMs) in real-world settings.
> > > >
> > > > While throughput-focused evaluations are valuable for general-purpose scenarios, our focus lies elsewhere. Our primary objective is to ensure functionality and reliability in extreme memory-constrained settings where existing methods, such as KIVI, might encounter out-of-memory (OOM) errors. KV-Dict provides a robust solution in scenarios where even a batch size of 1 exceeds memory limits with current compression approaches. We believe that pushing the Pareto frontier forward in such settings is a significant contribution.
> > > >
> > > > We acknowledge the latency trade-off in our design. By prioritizing memory efficiency, KV-Dict sacrifices some latency to operate in scenarios where existing methods fail due to memory constraints. The prefill latency scales linearly with input length and is indeed slower than other compression methods. However, we note that reusable prompts-such as system prompts or static initialization sequences-can be pre-compressed and cached. In tasks like document QA, retrieval, multi-turn dialogue, or few-shot learning, the longer prefill time has minimal impact on practical usability.
> > > >
> > > > > seems like we need to add both the encoding and reconstruction parts in Table 5. which nearly doubles the latency.
> > > >
> > > > The compression and reconstruction processes can theoretically be performed in parallel. Under this assumption, the theoretical lower bound for decoding latency in Table 5 would be 56.35 ms. However, the current implementation does not support this parallel computation, resulting in these two stages being processed sequentially. Despite this limitation, we have further optimized the OMP algorithm, reducing the decoding latency to 79.91 ms per token.

---

> > > > > ### Author Response · Authors · 2024-12-03
> > > > > **Official Comment by Authors**
> > > > >
> > > > > We would like to kindly follow up regarding our responses to your feedback on our submission. We deeply appreciate the time and effort you have invested in reviewing our work, and we have worked diligently to address your concerns during the rebuttal phase. With the rebuttal period nearing its end, we would greatly appreciate your thoughts on whether our updates have adequately addressed your concerns.
> > > > >
> > > > > To recap, we have made several key updates:
> > > > >
> > > > > * **Expanded Baselines:** Added comparisons with four additional KV cache compression methods (ZipCache, SnapKV, PyramidKV, and Per-Token Quantization) with results presented as Pareto curves.
> > > > > * **Additional model sizes and tasks:** Included results for new model sizes (1B and 3B) and challenging tasks (e.g., MMLU-Pro Engineering/Law).
> > > > > * **Clarified Methodology:** Revised the introduction and methodology to better communicate the motivation and novelty of KV-Dict.
> > > > >
> > > > > We are confident that the performance of KV-Dict demonstrates significant improvements, particularly in memory-constrained scenarios. While the latency trade-off is a noted weakness, we prioritize addressing extreme memory limitations, which are increasingly critical in real-world deployments. KV-Dict enables functionality in scenarios where existing methods face out-of-memory issues. Despite the higher latency, we have optimized our implementation to reduce decoding time and outlined future directions for further improvements.
> > > > >
> > > > > Given the comprehensive updates and additional experiments, we would be grateful if you could share your perspective on whether these revisions address your concerns and justify a potential score re-evaluation. If there are any remaining questions or suggestions, we would be more than happy to provide further clarifications promptly.

---

> ### Author Response · Authors · 2024-11-23
>
> Thank you for your valuable feedback on our paper. We have provided our responses to your comments and would greatly appreciate it if you could kindly review them. Please let us know if there are any additional points we can clarify to assist in your evaluation.

---

### Official Review · Reviewer_Uzgm · 2024-11-03

**Soundness:** 3
**Presentation:** 2
**Contribution:** 2
**Rating:** 5
**Confidence:** 3

**Summary:**

The KV cache of Large Language Models (LLMs) scales in proportion to both the model size and input sequence length, resulting in increased GPU memory usage. This poses significant constraints when deploying LLMs. To address this limitation, this paper proposes a novel method called KV-Dict, which compresses the KV cache efficiently. KV-Dict leverages sparse dictionary learning to reconstruct the KV cache in a near-lossless manner using a small calibration dataset. For creating sparse representations, it employs Orthogonal Matching Pursuit (OMP), and the dictionary is directly learned through gradient updates. The proposed method demonstrates higher accuracy compared to existing approaches and maintains satisfactory accuracy even at higher compression ratios than 2-bit quantization.

**Strengths:**

- The proposed method exhibits higher accuracy than existing KV cache compression methods at equivalent compression rates, demonstrating its superiority in preserving model performance under constrained memory settings.
- It achieves a higher compression rate than 2-bit quantization while maintaining robust and dependable accuracy, indicating its resilience even under significant compression.
- Despite performing compression using the Wikitext dataset, the method achieves commendable accuracy on the GSM8K dataset, illustrating its adaptability and generalization across different data types.

**Weaknesses:**

- Absence of throughput measurements: Given that a primary motivation for KV cache compression is to reduce GPU memory consumption, thereby enabling an increase in batch size and improving throughput, the paper would benefit from comparative throughput results against methods such as KIVI. This would provide a clearer picture of the practical benefits of KV-Dict in real-world deployments.
- Increased latency: Although this method achieves impressive compression rates and accuracy, it does so at the cost of increased latency, which may impact the feasibility of its deployment in latency-sensitive applications.
- Limited experimental validation across varying model sizes: The majority of experiments have been conducted on models in the 7B/8B range. Additional validation on both larger and smaller models would strengthen the findings, particularly to assess whether the dictionary exhibits universal properties across diverse model sizes.

**Questions:**

- What is the end-to-end latency? While the paper provides latency measurements for each component in Table 5 and notes that minor overhead is introduced due to parallel processing, an actual end-to-end latency measurement would be beneficial to understand the full extent of this overhead in practice.
- What batch size was used in the latency measurements?

---

> ### Author Response · Authors · 2024-11-20
> **Authors' response to Reviewer Uzgm (R3)**
>
> We thank the reviewer for recognizing the strengths of our work and for providing helpful suggestions and comments.
>
> ### W3: Varying model sizes
>
> We have conducted additional experiments on 1B and 3B models and present our findings in our General Response "Q2: Additional experiments comparing more compression methods, model sizes, and tasks". Our results demonstrate that KV-Dict remains effective across diverse model sizes, and its universal property holds consistently. This is shown through comparisons with five KV cache compression methods: Per-Token Quantization, KIVI, ZipCache, SnapKV, and PyramidKV.
>
>
>
> ### W1&2,Q1&2: Latency and throughput measurements
> We thank the reviewer for highlighting concerns about latency and recognize its importance in certain deployment scenarios. However, we would like to clarify that the primary focus of our method is addressing highly memory-constrained environments, which are increasingly relevant for real-world deployments of large language models (LLMs). This is reflected in the growing interest in compression techniques [2, 3] and open-source tools such as [BitNet](https://github.com/microsoft/BitNet). In such scenarios, even a modest batch size of 1 can surpass memory limitations with existing baseline compression methods.
>
> By contrast, KV-Dict avoids OOM errors by achieving significantly more efficient KV cache compression. This advantage is substantiated by our additional experiments spanning diverse model sizes, tasks, and baselines, demonstrating the method’s robustness and adaptability (see Figure 4 and 5). While this comes with increased latency, we believe the ability to ensure reliable performance under stringent memory constraints is a critical contribution. Notably, latency trade-off is common in KV cache compression, similarly to the classic space-time trade-offs in computer science. For instance, KIVI also exhibits increased latency, as shown in [Figure 4](https://openreview.net/pdf?id=L057s2Rq8O) in [1].
>
> We acknowledge the reviewer’s concern that higher latency may be a limitation for latency-critical use cases. While our focus has been on optimizing for memory efficiency rather than latency, we include end-to-end latency measurements for prefilling and decoding: with an input length of 1000 tokens and a batch size of 1, the standard implementation takes 0.14 seconds to prefill and 48.39 ms/token for decoding; KIVI takes 0.15 seconds to prefill and 59.66 ms/token for decoding; and KV-Dict takes 6.98 seconds to prefill and 96.93 ms/token for decoding. Note that implementing more sophisticated approximation algorithms or hardware optimizations (as KIVI does) may yield significant improvement in latency.
>
> In summary, our method addresses a critical challenge in memory-constrained LLM deployments with limitations of increased prefilling latency, offering a valuable contribution to pushing the pareto frontier in highly memory-constrained settings.
>
>
>
> ### References
>
> [1] Liu, Z., Yuan, J., Jin, H., Zhong, S., Xu, Z., Braverman, V., Chen, B. and Hu, X., KIVI: A Tuning-Free Asymmetric 2bit Quantization for KV Cache. In Forty-first International Conference on Machine Learning.
>
> [2] Luohe, S., Zhang, H., Yao, Y. and Li, Z., Keep the Cost Down: A Review on Methods to Optimize LLM’s KV-Cache Consumption. In First Conference on Language Modeling.
>
> [3] Ma, S., Wang, H., Ma, L., Wang, L., Wang, W., Huang, S., Dong, L., Wang, R., Xue, J. and Wei, F., 2024. The era of 1-bit llms: All large language models are in 1.58 bits. arXiv preprint arXiv:2402.17764.

---

> ### Author Response · Authors · 2024-11-23
>
> Thank you for your valuable feedback on our paper. We have provided our responses to your comments and would greatly appreciate it if you could kindly review them. Please let us know if there are any additional points we can clarify to assist in your evaluation.

---

> > ### Comment · Reviewer_Uzgm · 2024-11-24
> >
> > Thank you for the detailed rebuttal and additional experiments, which provide valuable insights into KV-Dict's performance and clarify its focus on memory efficiency. However, my primary concern regarding the significant latency overhead remains. As the latency-performance trade-off is critical for real-world applications and no concrete solutions are presented to address this, I will maintain my original score.

---

### Official Review · Reviewer_p9ML · 2024-11-03

**Soundness:** 3
**Presentation:** 2
**Contribution:** 3
**Rating:** 6
**Confidence:** 3

**Summary:**

The paper introduces KV-Dict, a method for compressing the key-value (KV) cache in Transformer-based Large Language Models (LLMs) to address the significant memory demands during inference caused by the KV cache growing proportionally with model size and input context length. KV-Dict leverages a small, universal dictionary as an overcomplete basis to sparsely decompose and accurately reconstruct KV states using sparse dictionary learning techniques, specifically Orthogonal Matching Pursuit (OMP). This approach captures low-dimensional structures within the KV cache, allowing for significant compression (3-8×) with minimal reconstruction error. By providing fine-grained control over sparsity levels, KV-Dict enables users to balance memory savings against performance loss, achieving near-lossless performance even in low-memory regimes where traditional quantization methods struggle. The method is universal and input-agnostic, simplifying deployment across different tasks and batches without retraining or scaling memory. KV-Dict maintains strong performance on complex reasoning tasks like LongBench and GSM8K, offering an efficient, off-the-shelf solution for pre-trained LLMs to alleviate the KV cache bottleneck by exploiting redundancy and low-dimensional structures in KV states.

**Strengths:**

- There is a comprehensive set of experiments and ablations to validate the hypothesis and methods presented in this work.
- The problem is relevant and very practical. KV-Cache compression is a key bottleneck for LLM development.
- Speed and performance is competitive across a wide suite of benchmarks

**Weaknesses:**

### Major
- Writing: The writing in this paper is hard to follow and at some points it is hard to understand the motivation for why things are being done. After reading the abstract for example, I am unable to tell wether this is a sparsity based method or quantization method or both. There are also a bunch of grammatical errors littered around the paper that make it hard to read e.g "such dictionary does not scale" line 095.
- Besides, performance on benchmarks it would be great to see some qualitative results with samples to see what the output from LLMs using KV-Dict look like compared to those from a few other methods or when using the full KV-cache without compression.

**Questions:**

N/A

---

> ### Author Response · Authors · 2024-11-20
> **Authors' response to Reviewer p9ML (R2)**
>
> ### W1: Presentation is hard to follow
> We thank the reviewer for pointing out ways to improve our presentation. We have revised our manuscript to include the following edits to address the reviewer's presentation concerns:
> - L80-94: In our updated introduction, we have included a summary of the end-to-end KV-Dict pipeline in three steps. Here, we clarify that we train our dictionary using WikiText-103 and apply it to all tasks, which was ambiguous to a couple reviewers. With this summary, we clearly separate our method and contributions, which follow in L96-107.
> - L82-83: The method summary explains that we use both sparsity-based compression and quantization. However, the 8-bit quantization is a minor detail that boosts our performance by reducing memory storage of sparse coefficients.
> - L289-291: We include omitted details of dictionary pretraining in the Experiments section.
> - Figure 4&5: For clearer result presentation, we include multiple plots that demonstrate KV-Dict versus other baseline methods. We plot KV cache size vs. benchmark performance and demonstrate the Pareto frontier of each method.
> - We have fixed some grammatical errors as well, though we did not highlight these in red for the sake of readability.
>
>
>
> ### W2: Qualitative results of output samples
> As requested, below we provide an example generation for qualitative analysis. We present the original GSM8K problem and solution below, along with outputs from Llama-3.1-8B-Instruct with full KV cache (FP16), KV-Dict (sparsity $s=14$), and KIVI-2. In the example, we see that KV-Dict and KIVI-2 first performs a useless calculation that 3 German Shepherds and 2 bulldogs are 5 dogs. However, KV-Dict gets back on track and generates the correct solution, while KIVI-2 continues on to incorrect reasoning.
>
> We have attached the example in page 16-17, Appendix D.
>
> ---
> **Problem:** The Kennel house keeps 3 German Shepherds and 2 Bulldogs. If a German Shepherd consumes 5 kilograms of dog food and a bulldog consumes 3 kilograms of dog food per day. How many kilograms of dog food will they need in a week?
>
> **Solution:** The German Shepherds consume 3 x 5 = <<3\*5=15>>15 kilograms of dog food per day.
> The Bulldogs consume 2 x 3 = <<2\*3=6>>6 kilograms of dog food per day.
> The dogs can consume a total of 15 + 6 = <<15+6=21>>21 kilograms of dog food per day.
>
>
> Therefore, they need 21 x 7 = <<21\*7=147>>147 kilograms of dog food in a week.
>
> **Final Answer:** 147
>
> ---
> **Llama 8B with Full Cache:**
>
>  First, find the total amount of food the German Shepherds consume in a day: 3 dogs * 5 kg/dog = 15 kg
>
>
> Then find the total amount of food the bulldogs consume in a day: 2 dogs * 3 kg/dog = 6 kg
>
>
> Then add the two amounts together to find the total amount of food the dogs consume in a day: 15 kg + 6 kg = 21 kg
>
>
> Then multiply that number by 7 to find the total amount of food the dogs consume in a week: 21 kg/day * 7 days = 147 kg
>
> ---
> **Llama 8B with KV-Dict (s=14):**
>
> The kennel has 3 German Shepherds and 2 bulldogs, so they have 3 + 2 = 5 dogs.
>
>
>
>
> The German Shepherds consume 5 * 3 = 15 kilograms of dog food per day.
>
>
> The bulldogs consume 2 * 3 = 6 kilograms of dog food per day.
>
>
> So, the kennel needs 15 + 6 = 21 kilograms of dog food per day.
>
>
> In a week, they will need 21 * 7 = 147 kilograms of dog food.
>
>
> The answer is 147.
>
> ---
> **Llama 8B with KIVI-2:**
>
>  The Kennel house has 3 German Shepherds and 2 Bulldogs, so they have a total of 3 + 2 = <<3+2=5>>5 dogs.
>
> Each German Shepherd consumes 5 kilograms of dog food per day, so the total amount of dog food consumed by the German Shepherds is 5 * 5 = <<5\*5=25>>25 kilograms per day.
>
> Each bulldog consumes 3 kilograms of dog food per day, so the total amount of dog food consumed by the bulldogs is 2 * 3 = <<2\*3=6>>6 kilograms per day.
>
> The total amount of dog food consumed per day is 25 + 6 = <<25+6=31>>31 kilograms.
>
> The Kennel house will need 31 * 7 = <<31\*7=217>>217 kilograms of dog food in a week.

---

> > ### Comment · Reviewer_p9ML · 2024-11-22
> >
> > Thanks to the authors for the detailed reply. I will keep my score.

---

### Official Review · Reviewer_ArJZ · 2024-11-03

**Soundness:** 3
**Presentation:** 3
**Contribution:** 2
**Rating:** 5
**Confidence:** 4

**Summary:**

The paper is on the challenge of deploying large language models (LLMs) – the memory consumption required to manage the key-value (KV) cache during inference. The paper proposes a method, KV-Dict, which uses a sparse dictionary-based approach for post-training KV cache compression.

**Strengths:**

The paper introduces sparse dictionary learning as a method to approximate KV cache compression, which seems to be a new contribution.
The KV-Dict method demonstrates comparable performance vs. a recent study KIVI.

**Weaknesses:**

- The study primarily compares KV-Dict with KIVI, which, while valuable, may not provide a comprehensive view of its effectiveness relative to other potential KV cache compression techniques. Additional comparisons with more methods would strengthen the validity of the approach and its claims.
- Although KV-Dict is claimed to be universal across different inputs, the paper could benefit from a more detailed explanation and empirical studies of this claim. Demonstrating the effectiveness across a broader range of tasks and input distributions would help substantiate this claim.
- While the approach is technically sound, it may not introduce significantly new dimensions to the KV cache compression problem beyond the use of sparse dictionary learning. Highlighting new aspects of KV cache compression could enhance the paper’s contribution.

**Questions:**

- What was the primary motivation behind designing KV-Dict as a universal dictionary rather than task-specific dictionaries? Is this approach primarily for memory savings, or are there anticipated advantages in model generalizability?
- For a model such as Llama-3-8B, could the authors detail the procedure for model serving with KV-Dict? The paper mentions that dictionary learning was performed using WikiText-103. Is this same dictionary applied directly during prefilling and decoding stages, or would additional fine-tuning be necessary?
- Could the authors clarify the mechanism for obtaining compressed Key and Value states at model serving time? Is the dictionary pre-computed and directly used, or is there a dynamic adaptation of the dictionary as new inputs are processed?

---

> ### Author Response · Authors · 2024-11-20
> **Authors' response to Reviewer ArJZ (part 1)**
>
> We thank the reviewer for recognizing the contribution of our work, and for the helpful suggestions and comments. Below we address the suggestions and questions by the reviewer ArJZ.
>
> ### W1: Additional comparison with more KV cache compression methods
> Please refer to our General Response, where we address this question in our response to "Q2: Additional experiments comparing more compression methods, model sizes, and tasks".
>
>
> ### W2: Broader range of tasks and input distributions
> First, we highlight that we originally evaluated LongBench, which covers 8 different tasks ranging from summarization of news articles (MultiNews) to code completion (LCC, RepoBench-P). This already includes a diverse set of benchmarks and word distributions, demonstrating KV-Dict's generalizability.
>
> Additionally, we conducted experiments on the MMLU-Pro dataset (using our universal distionary trained on WikiText-103) to test its performance on an even broader variety of input distributions. Specifically, we chose to evaluate two subjects with very different word distributions: Engineering and Law. Please refer to our General Response "Q2: Additional experiments comparing more compression methods, model sizes, and tasks" for our results and analysis. The results substantiate our claim that our dictionary is generalizable across input distribution.
>
>
> ### W3: Does not introduce new dimensions to KV cache compression
> We disagree with the reviewer regarding novelty for two reasons. We believe that not only is our method new in its implementation, but also that our results are quantitatively novel.
> 1. KV-Dict is the first approach to decompose KV cache into a basis and incorporate dictionary learning, to the best of our knowledge. Furthermore, our hybrid algorithm to train the dictionary using OMP as an "encoder" and perform gradient descent on the decoder linear layer (*i.e.*, dictionary) differentiates our method from prior sparse autoencoder literature.
> 2. Quantitatively, we venture into low-memory regimes in which previous works have been unable to tread; mainly the fact that KV-Dict can employ higher compression rates than 2-bit quantization can offer (L101-104). Both our original and additional experiments support this claim by outperforming quantization and eviction methods across model sizes and multiple benchmarks.

---

> > ### Author Response · Authors · 2024-11-20
> > **Authors' response to Reviewer ArJZ (part 2)**
> >
> > ### Q1: Motivation behind designing a universal dictionary instead of task-specific dictionaries
> > (Please refer to our General Response, where we partially address this question in our response to "Q1: How to initialize and train KV-Dict".)
> >
> > Our approach allows the universal dictionary to be used off-the-shelf for various tasks without the need for additional training, making it highly versatile and efficient. That said, we can see that one may want to explore task-specific dictionaries to further enhance the performance of KV-Dict in specific domains.
> >
> > Below, we present experimental results with the following dictionaries:
> > 1. Universal dictionary: Dictionary trained on WikiText-103.
> > 2. Dictionary pre-trained on GSM8K: Dictionary trained from scratch on the GSM8K training set.
> > 3. Universal dictionary finetuned on GSM8K: Dictionary trained on WikiText-103 and then fine-tuned on GSM8K.
> >
> > | Sparsity | KV size | Universal dictionary | Dictionary pre-trained on GSM8K | Universal dictionary finetuned on GSM8K |
> > | -------- | -------- | -------- | -- | -- |
> > | $s = 24$     | 36.9%     | 76.88     | 77.63 | 79.68 |
> > | $s = 14$     | 26.1%     | 75.06     | 70.51 | 75.89 |
> > | $s = 10$     | 22.7%     | 69.67     | 59.59	| 67.25 |
> >
> > From the experimental results, we observe that training a dictionary from scratch on GSM8K significantly underperforms compared to the universal dictionary, especially at low sparsities. This indicates overfitting to the small GSM8K training set, failing to generalize even to its test set. In contrast, fine-tuning a universal dictionary on GSM8K provides marginal improvements but adds the additional overhead of custom fine-tuning. The slight performance gain may not justify the additional complexity and computational cost required for fine-tuning, especially when the universal dictionary already offers strong performance with simplicity and efficiency.
> >
> >
> > ### Q2&3: Clarification on methodology
> > Yes, the dictionary learned using WikiText-103 (as described in Section 2.3) is the dictionary used throughout all experiments for both prefilling and decoding, without any fine-tuning. At model serving time, the pre-computed universal dictionary remains fixed. For specific details on how the dictionary is initialized and trained, please refer to our General Response, where we address this question in our response to "Q1: How to initialize and train KV-Dict".
> >
> > Compressed key and value states are obtained by leveraging this pre-computed dictionary. Specifically, during the prefilling stage, the OMP algorithm computes the sparse representations of the KV states using the dictionary; in other words, OMP finds the closest linear combination using only $s$ many vectors of the dictionary as a basis. These compressed representations are then cached. During the decoding stage, the same dictionary is utilized to compress new KV states as needed. For further clarity, we have updated the intro with an overview of the KV-Dict method (L82-94).
> >
> > Though secondary to our main findings, in Appendix C.3, we also experimented with a variant of our method, which adaptively learns the dictionary throughout the generation process. This approach begins with an even smaller dictionary and dynamically adds KV states that are not approximated well enough. While this improves performance compared to the universal dictionary baseline, the additional dictionary atoms vary across different inputs and require extra memory proportional to the batch size. Consequently, despite the performance improvement, the method does not achieve a better performance on the Pareto frontier (Figure 4 and 5) due to the increased memory requirement.

---

> > > ### Author Response · Authors · 2024-11-23
> > >
> > > Thank you for your valuable feedback on our paper. We have provided our responses to your comments and would greatly appreciate it if you could kindly review them. Please let us know if there are any additional points we can clarify to assist in your evaluation.

---

> > > > ### Comment · Reviewer_ArJZ · 2024-11-26
> > > >
> > > > Thank you for your response and for providing additional comparisons. I appreciate the effort in addressing the points raised. While the new results are helpful, I still have concerns regarding the practical applicability of the proposed method. The use of sparse dictionary learning for KV cache compression is indeed a promising direction and could have significant impact. However, the system performance and implementation details remain unclear, which could be strengthened when comparing with other methods. Moreover, the claim that the proposed method is novel in low-memory regimes where prior work falls short could be further substantiated, given that other methods can achieve the similar compression ratios. Overall, while I acknowledge the additional efforts made in the response, I will maintain my original rating.

---

> > > > > ### Author Response · Authors · 2024-11-27
> > > > > **Authors' response to Reviewer ArJZ**
> > > > >
> > > > > Thank you for reviewing our updated submission and rebuttal. We address the reviewer's comments below:
> > > > >
> > > > > > the system performance and implementation details remain unclear, which could be strengthened when comparing with other methods.
> > > > >
> > > > > We would like to better understand which specific aspects of system performance and implementation you find unclear. To provide additional clarity, we have included the pseudocode of our algorithm in Appendix A (Algorithm 2). Additionally, we have shared our preliminary codebase, which is available at: https://anonymous.4open.science/r/KV-Dict-8F25/.
> > > > > We recognize that, due to time constraints, the code is not yet fully polished. However, we are committed to releasing a refined version publicly in the future.
> > > > > In the meantime, we would greatly appreciate any specific suggestions on how we might improve the presentation or provide further details to better address your concerns.
> > > > >
> > > > > > the claim that the proposed method is novel in low-memory regimes where prior work falls short could be further substantiated, given that other methods can achieve the similar compression ratios.
> > > > >
> > > > > We would greatly value recommendations of these other methods. As far as we are aware, our experiments represent the best memory efficiency achievable with state-of-the-art approaches (PyramidKV, SnapKV, KIVI, ZipCache). While eviction-based methods can theoretically reduce memory usage arbitrarily, this typically results in substantial performance degradation, as demonstrated by the significant drop in performance at 65% KV cache for Llama-3.1-8B-Instruct on GSM8K. Similarly, for quantization methods, even the most memory-saving implementations typically require at least 2 bits, which imposes a clear upper bound on achievable compression rates. Please refer to Figure 4 where we have tested each baseline at its *highest* possible compression rate.
> > > > >
> > > > > If there are prior works that achieve higher compression rates, we would be grateful if the reviewer could point us to them. This would allow us to incorporate these methods into our comparisons and further strengthen the robustness of our claims.

---

> > > > > > ### Author Response · Authors · 2024-12-03
> > > > > > **Official Comment by Authors**
> > > > > >
> > > > > > We would like to kindly follow up regarding our responses to your feedback on our submission. We have taken great care to address your concerns through additional experiments, comparisons, and improvements to the manuscript. As the rebuttal period is nearing its close, we would greatly value any further comments or suggestions you might have.
> > > > > >
> > > > > > To recap we have:
> > > > > >
> > > > > > * Conducted additional comparison with four more KV cache compression methods including both state-of-the-art quantization and eviction methods.
> > > > > > * Evaluated KV-Dict's robustness across a broader range of tasks to further substantiate our claims.
> > > > > > * Performed experiments with task-specific dictionaries based on the reviewer's suggestions.
> > > > > > * Significantly revised our manuscript to clarify the methodology and address potential ambiguities.
> > > > > >
> > > > > > We hope these updates adequately address your concerns and would greatly appreciate it if you could share any remaining suggestions or feedback at your earliest convenience. Additionally, we would be grateful if you could let us know whether these revisions justify a potential score re-evaluation.

---

### Author Response · Authors · 2024-11-20
**General Response to AC and Reviewers**

(**R1** = R-ArJZ, **R2** = R-p9ML, **R3** = R-Uzgm, **R4** = R-XfrT)

We sincerely appreciate the thoughtful and constructive feedback from the reviewers. We are encouraged by the positive remarks from all reviewers, acknowledging the methodological novelty of KV-Dict (**R1**, **R3**, **R4**), and the effectiveness of our method under memory-constrained regimes (**R1**, **R2**, **R3**), a setting that is of practical concern for LLM development (**R2**). We are pleased that reviewers have identified contributions beyond performance gain. In particular, we design a universal dictionary that is adaptable and generalizable across different data types (**R3**), while providing a comprehensive set of experiments and ablations to investigate our method's strengths and weaknesses (**R2**).

We have uploaded a newly revised manuscript, which includes a revised introduction that clarifies the KV-Dict method (L82-94) and additional experimental results with multiple different baselines and language models (L373-431). We believe our new experiments show our method stands out among other compression schemes and would be of interest to reviewers.

We respond to two common questions here in the general response. Other individual questions and comments from the reviewers are addressed separately in each reply.


## Q1: How to initialize and train KV-Dict (R1, R4)
In general, we design KV-Dict as a universal dictionary to prioritize generalizability and ease of access. By training the dictionary on a large, diverse dataset such as WikiText, we aim to generalize effectively across a wide range of input distributions. This approach allows the universal dictionary to be used off the shelf for various tasks without the need for additional training, making it highly versatile and efficient. We then evaluate our method's performance on GSM8K, MMLU-Pro (Engineering, Law), and LongBench (eight benchmarks), which spans multiple topics and word distributions.

To achieve this, all dictionaries used in our experiments were only trained with WikiText-103, except in our ablation study that employs an adaptive dictionary (L468-474 and Appendix C.3). The dictionaries are initialized with a uniform distribution, following the default initialization method for linear layers in PyTorch. We make this clearer in L214-215 and L289-291 of our revised manuscript.

---

> ### Author Response · Authors · 2024-11-20
> **General Response to AC and Reviewers (continued)**
>
> ## Q2: Supplemental experiments comparing more compression methods, model sizes, and tasks (R1, R2, R3, R4)
>
> We appreciate the reviewers' suggestions to strengthen our claims. In response, we provide three dimensions in which we strengthen our results: comparing against more compression methods, testing more model sizes, and evaluating on additional tasks.
>
> ### Experimental Setup
> **Compression baselines.** We conduct extensive experiments and provide comparisons with four new KV cache compression methods: Per-Token Quantization, ZipCache [2], SnapKV [3], and PyramidKV [1]. Originally, we have only compared KV-Dict against KIVI.
>
> - Per-Token Quantization refers to the conventional token-wise quantization approach.
> - KIVI is a recent asymmetric quantization method, which quantizes keys per-channel and values per-token in groups of size $g$.
> - ZipCache employs mixed-precision quantization and achieves state-of-the-art performance in the category of quantization-based compression.
> - SnapKV and PyramidKV are state-of-the-art eviction-based compression methods.
>
> Since these methods achieve varying compression ratios, we present our results using a Pareto curve. The x-axis represents the KV cache size, while the y-axis represents benchmark performance. This visualization enables a clear comparison of trade-offs across methods (*e.g.*, see Figure 4 in our revised submission).
>
> **Model sizes.** We also add two new model sizes: Llama-3.2-1B-Instruct and Llama-3.2-3B-Instruct. Previously, we have examined the models Llama-3-8B, Llama-3.1-8B-Instruct, and Mistral-7B-Instruct-v0.3.
>
> **Additional tasks.** For all new baselines and new models, we test GSM8K, MMLU-Pro Engineering, and MMLU-Pro Law across multiple KV cache compression rates. We choose the MMLU-Pro subjects Engineering and Law due to their difficulty, as explained in [4]. We choose more difficult subjects because we tend to see stronger differentiation and complex reasoning is often sensitive KV cache compression loss.
>
>
> ### Results and Analysis
> Our results demonstrate that KV-Dict often achieves stronger performance than all other baseline methods on GSM8K and MMLU-Pro. For full comprehensive results, please refer to Figures 4 and 5 in our revised paper for Pareto curves for each model and task.
>
> **GSM8K Results across model sizes**
> | Model | Full cache (fp16) | KV-Dict | ZipCache | KIVI | Per-Token Q | PyramidKV | SnapKV |
> |-------|------------|---------|----------|------|-------------|-----------|--------|
> | Llama-3.2-1B-Instruct | 35.78 (100%) | **33.51** (25.4%) | 24.11 (28.2%) | 14.78 (25.7%) |  1.21 (25.7%) |  2.50 (67.2%)  |  5.00 (66.3%)  |
> | Llama-3.2-3B-Instruct | 66.11 (100%) | **65.81** (26.1%) | 61.33 (28.2%) | 53.53 (25.7%) | 14.56 (25.7%) | 13.80 (49.8%) | 12.74 (48.9%) |
> | Llama-3.1-8B-Instruct | 77.26 (100%) | **75.06** (26.1%) | 73.54 (28.2%) | 69.60 (25.7%) | 28.35 (25.7%) | 14.78 (68.0%) | 16.30 (67.4%) |
>
>
> Not only have we originally shown KV-Dict excelling on GSM8K in Table 3, but we have now painted a more complete picture by running experiments across a spectrum of KV cache sizes and baselines. The advantage holds true for all across model sizes, as seen in our results with Llama-3.2-1B-Instruct and Llama-3.2-3B-Instruct on GSM8K.
>
> We also observe that eviction-based methods significantly underperform. This is because PyramidKV and SnapKV are not compatible with Grouped Query Attention (GQA), an architectural component that uses fewer key-value attention heads and frequently used in recent models (*e.g.*, Llama 3, Mistral v0.3, Qwen 2).
>
>
> **MMLU-Pro Results with Llama-3.1-8B-Instruct**
> | Subject | Full cache (fp16) | KV-Dict | ZipCache | KIVI | Per-Token Q | PyramidKV | SnapKV |
> |---------|------------|---------|----------|------|-------------|-----------|--------|
> | Engineering | 24.05 (100%) | **22.09** (26.9%) | 20.64 (27.3%) | 20.43 (24.7%) | 15.38 (24.7%) |  0.10 (45.6%) | 0.52 (45.4%) |
> | Law         | 27.70 (100%) | 27.07 (28.1%) | 27.79 (26.1%) | **28.97** (24.3%) | 21.16 (24.3%) | 1.45 (39.9%) | 3.54 (39.8%) |
>
> When evaluating KV-Dict on MMLU-Pro, we observe similar gains, especially compared to eviction-based methods PyramidKV and SnapKV. One caveat is that we see our performance is slightly worse than ZipCache and KIVI when KV cache size is around 25%. However, in low memory regime below 20% cache, our method still outperforms any other baseline. This highlights that KV-Dict supports a wide range of compression ratios quite effectively and that our dictionary is generalizable across input distribution.

---

> > ### Author Response · Authors · 2024-11-20
> > **General Response to AC and Reviewers (continued; part 2)**
> >
> > ### Conclusion
> > In our paper, we claim two main contributions in terms of performance: near lossless compression given similar memory constraints to 2 or 4 bit quantizations (L98-100) and compression rates beyond 2 bits (L101-104). Our supplemental experiments further support our claims by outperforming quantization and eviction methods across model sizes and multiple benchmarks. Even in memory regimes in which ZipCache or KIVI achieve slightly better accuracies, our method does not deviate much from full cache performance.
> >
> > ### References
> > [1] Cai, Z., Zhang, Y., Gao, B., Liu, Y., Liu, T., Lu, K., Xiong, W., Dong, Y., Chang, B., Hu, J. and Xiao, W., 2024. Pyramidkv: Dynamic kv cache compression based on pyramidal information funneling. arXiv preprint arXiv:2406.02069.
> >
> > [2] He, Y., Zhang, L., Wu, W., Liu, J., Zhou, H. and Zhuang, B., 2024. ZipCache: Accurate and Efficient KV Cache Quantization with Salient Token Identification. arXiv preprint arXiv:2405.14256.
> >
> > [3] Li, Y., Huang, Y., Yang, B., Venkitesh, B., Locatelli, A., Ye, H., Cai, T., Lewis, P. and Chen, D., 2024. Snapkv: Llm knows what you are looking for before generation. arXiv preprint arXiv:2404.14469.
> >
> > [4] Wang, Y., Ma, X., Zhang, G., Ni, Y., Chandra, A., Guo, S., Ren, W., Arulraj, A., He, X., Jiang, Z. and Li, T., 2024. Mmlu-pro: A more robust and challenging multi-task language understanding benchmark. arXiv preprint arXiv:2406.01574.

---

### Meta-Review · Area_Chair_1UUV · 2024-12-20

**Metareview:**

This paper proposes KV-Dict for compressing LLM key-value caches, offering strong memory savings with minimal accuracy drop. While novel, the method’s latency overhead and limited clarity raise concerns. The authors addressed some issues with more experiments and clarifications, but the main latenncy problem remains. Reviewers still question the practical impact and universality claims.

Without sufficient improvement in latency and throughput analysis, the paper remains borderline and not quite ready for publication. Better clarity, a thorough latency-performance tradeoff evaluation, and more extensive baselines would help. This work is borderline, but the final decision is to reject this work.

**Additional Comments On Reviewer Discussion:**

The reviewers initially raised three main concerns: 1) limited baseline comparisons, 2) latency overhead, and 3) unclear methodology/universality claims. The authors addressed these by adding comparisons with four new compression methods, testing additional model sizes (1B/3B), and evaluating on more tasks like MMLU-Pro. While the expanded experiments demonstrated stronger performance gains across different settings, particularly for memory-constrained scenarios, concerns about latency overhead remained unresolved. Most reviewers maintained their original scores, suggesting the responses adequately clarified the method's strengths but did not fully address practical deployment concerns.

---

### Decision · Program_Chairs · 2025-01-22

Reject